# Histone exchange sensors reveal variant specific dynamics in mouse embryonic stem cells

Marko Dunjić[1,3], Felix Jonas[2,3], Gilad Yaakov [ID][2,3], Roye More [ID][1], Yoav Mayshar[1], Yoach Rais[1], Ayelet-Hashahar Orenbuch[1], Saifeng Cheng[1], Naama Barkai [ID][2] & Yonatan Stelzer [ID][1] ✉

Eviction of histones from nucleosomes and their exchange with newly synthesized or alternative variants is a central epigenetic determinant. Here, we define the genome-wide occupancy and exchange pattern of canonical and non-canonical histone variants in mouse embryonic stem cells by genetically encoded exchange sensors. While exchange of all measured variants scales with transcription, we describe variant-specific associations with transcription elongation and Polycomb binding. We found considerable exchange of H3.1 and H2B variants in heterochromatin and repeat elements, contrasting the occupancy and little exchange of H3.3 in these regions. This unexpected association between H3.3 occupancy and exchange of canonical variants is also evident in active promoters and enhancers, and further validated by reduced H3.1 dynamics following depletion of H3.3-specific chaperone, HIRA. Finally, analyzing transgenic mice harboring H3.1 or H3.3 sensors demonstrates the vast potential of this system for studying histone exchange and its impact on gene expression regulation in vivo.

Nucleosomes, DNA-wrapped histone octamers that contain two H2A-H2B heterodimers and an H3·H4 tetramer[1], are the main structural units of eukaryotic chromatin. In addition to their structural function in DNA compaction, nucleosome composition shapes the epigenomic landscape by affecting DNA accessibility to factors related to transcription, replication, repair, and heterochromatin formation[2,3]. Nucleosomes are also subjected to post-translational modifications (PTMs) on histone tails, deposited through specialized enzymes, which could define position-specific combinatorial codes[4,5]. Conversely, the replacement of the nucleosome subunits, termed histone exchange (also known as turnover), effectively resets position-dependent histone PTMs by replacing them with histones evicted elsewhere and carrying associated modifications, or with histones carrying general modifications added to yet unbound newly-synthesized histones[6–9]. Unlike the deposition of histone modifications, which is gradual and depends on the concerted activity of *trans* factors, histone exchange

resets all modifications carried by the associated histones, thus significantly shaping the epigenomic landscape[10].

Histone variants represent another key factor influencing local and genome-wide nucleosome organization. For example, the histone variant H2A.X is incorporated onto DNA damage sites, and multiple testis-specific histone variants dynamically regulate chromatin accessibility during spermatogenesis[11–17]. Nucleosome composition in turn affects the binding dynamics of distinct histone subunits in specific regions. For instance, H2A/H2B heterodimers exchange more readily as compared to H3/H4 tetramers[18–20]. Moreover, the replacement of core histones by intrinsically less stable histone variants reduces nucleosome stability and influences exchange rates[21]. Most histone variants are expressed during the S-phase of the cell cycle, providing the main supply of histones during replication, and are termed "canonical" (also known as core) variants. Others –known as non-canonical variants – are expressed throughout the cell cycle and differ

[1]Department of Molecular Cell Biology, Weizmann Institute of Science, 7610001 Rehovot, Israel. [2]Department of Molecular Genetics, Weizmann Institute of Science, 7610001 Rehovot, Israel. [3]These authors contributed equally: Marko Dunjić, Felix Jonas, and Gilad Yaakov. ✉e-mail: yonatan.stelzer@weizmann.ac.il

in protein structure and function from their canonical counterparts (reviewed in ref. [13]).

Locus-specific deposition of histones is mediated by associated chaperone complexes. For example, H3.3 is a widely studied non-canonical H3 variant that utilizes two distinct chaperone systems to incorporate into open chromatin and heterochromatin regions (HIRA, and ATRX/DAXX, respectively)[22,23]. In the context of histone exchange, H3.3 is currently the prime mammalian variant to be deeply characterized genome-wide, with high exchange shown to be associated with active promoters and distal regulatory regions[24–34]. Currently, a comprehensive mapping of the exchange landscape associated with other histone variants is largely lacking, mainly due to technical challenges in measuring exchange[25,33,35–38]. Such maps will not only provide a more detailed view of the mammalian epigenomic landscape, but also allow dissection of potential interactions between histone variants in regulating gene expression.

Most current methods used to study histone exchange implement pulse-chase labeling to detect the incorporation of newly synthesized histones as a proxy for histone exchange rates[24–27,30,33–38]. Such systems require multiple time-resolved samples, exhibit inherent measurement delays, and limit perturbation studies necessary for determining causal relationships between histone exchange, *trans*-acting factors, and local epigenetic changes. Here, we utilized a histone exchange sensor recently established in yeast that is modified at each genomic position, depending on local exchange rates. An attractive aspect of the sensor system is that it can separate between incorporation of histones (affecting histone occupancy) from their exchange rates, in which occupancy could remain unchanged, using a single measurement of an unperturbed sample at steady-state[39]. Implementing the sensor system in mouse embryonic stem cells (mESCs) facilitated detailed mapping of the exchange landscape associated with canonical (H3.1 and H2B) and non-canonical (H3.3) histone variants. In addition to high-exchange rates in open chromatin and transcribed regions, we found transcription-independent dynamics of core H3.1 histone variant at bivalent promoters. Replacement of H3.1 and H2B variants at repeat elements of heterochromatin regions is associated with H3.3 occupancy at these regions. In agreement with these findings, knockout of HIRA, a histone H3.3 specific chaperone at open chromatin, affected exchange levels of canonical histone H3.1 at enhancer and promoter regions. The experimental system described here can be broadly used to study histone exchange during development and in adult tissues.

## Results

### A sensor system for charting non-canonical histone H3.3 exchange in mESCs

To measure histone exchange dynamics in mESCs, we adapted a sensor system that has been established and validated in yeast[39]. The system is composed of a Tobacco Etch Virus (TEV) cleavage sequence placed between myc and HA tags (hereafter termed sensor) and fused to the C-terminus of histone variant of choice (Fig. 1a). The TEV protease is fused to the C-terminus of a complementary histone subunit. Of note, the formation of nucleosomes is a two-step process in which the assembly of the H3-H4 tetramer on DNA is followed by the binding of two H2A-H2B dimers. Therefore, tagged histone variants come into proximity only upon binding to DNA, where cleavage of the TEV recognition site and release of the myc tag can occur. In this framework, the dual tagging system allows computing endogenous histone occupancy and exchange dynamics. Specifically, the HA tag remains intact regardless of TEV cleavage and as such, reports on total histone occupancy. On the other hand, the myc tag depends on the co-residence time of the tagged subunits at the given locus. Long co-residence results in productive cleavage and is characterized by a low myc signal relative to HA. In contrast, a high myc signal indicates short residence time and hence rapid replacement of histones (Fig. 1a,b). The ratio between the levels of myc and HA, as

measured by chromatin immunoprecipitation sequencing (ChIP-seq), reports on region-specific levels of histone exchange (turnover). Apart from exchange, individual myc and HA tags provide information on alternative modes of histone dynamics, that is, histone incorporation and eviction (Fig. 1b). When new histones are incorporated into a free DNA, both myc and HA signals are expected to increase, whereas eviction of histones is characterized by reduced HA but invariant myc levels[39] (Fig. 1b). We note that we can only quantify relative (between different regions) but not absolute levels of new histone (myc) incorporation which may be low. In addition, the sensor system can detect the incorporation of new myc-tagged histones but cannot detect recycling of bound histones by untagged histones from the general pool.

To validate the system in mammalian cells, we first focused on the well-studied non-canonical H3.3 variant, which has been previously examined using pulse-chase methods[24–34]. To this end, we targeted a transgene stably expressing histone H3.3 fused to a sensor alongside H2B fused to the TEV protease into the *Hipp11* (*H11*) safe harbor locus in mESCs (Supplementary Fig. 1a). The TEV protease is preferably fused to canonical histones to ensure genome-wide coverage. We generated cell lines expressing alternative forms of the TEV-cleavage site that either allow or prevent cleavage, termed cleavable and non-cleavable (NC) versions, respectively. The latter also serves as a control for potential technical differences between myc and HA antibodies in ChIP experiments. Quantitative real-time PCR and western blotting validated the expression and proper cleavage activity of the sensor system (Fig. 1c and Supplementary Fig. 1b). Notably, comparing protein expression between the H3.3 sensor and native H3.3 histones detected very low levels of the cleavable form, suggesting technical limitations in detecting both variants within the same protein pool (Supplementary Fig. 1c). Importantly, transgene expression did not seem to affect the self-renewal and differentiation potential of mESCs, as demonstrated by spontaneous differentiation into embryoid bodies and robust contribution to mouse chimeras upon injection into host blastocyst (Supplementary Fig. 1d, e).

### Rapid exchange of histone H3.3 at transcription start sites and active enhancers

Genome-wide ChIP-seq analysis detected comparable read coverage of both myc and HA signals at distinct genomic regions in two independent experiments (Supplementary Fig. 1f). This analysis also confirmed a high correlation between HA and myc signals across the genome in cells expressing the non-cleavable sensor (Fig. 1d). We further detected high correlation between non-cleavable HA and myc signals within previously annotated histone H3.3 peaks[31], indicating that the tagged histones are properly incorporated into native nucleosomes (Fig. 1e and Supplementary Fig. 1g). Yet, analyzing the same regions in the cleavable sensor cell line, following internal normalization to the library size, identified different patterns of myc and HA signals. We observed a significant proportion of regions with relatively higher myc to HA ratios indicating rapid exchange, that partially overlap with transcription start sites (TSSs) of the top 1% of highly expressed genes (Fig. 1f). This observation prompted us to further explore the association between exchange patterns of histone H3.3 and gene transcription. Consistent with previous results[24–31,33,34], exchange levels around TSSs for protein-coding genes grouped by expression levels revealed a strong correlation between H3.3 exchange and RNA expression (Fig. 1g and Supplementary Fig. 1h). Notably, transcription also positively correlated with H3.3 occupancy, as demonstrated by increasing HA signal, confirming the high deposition of H3.3 at promoters of expressed genes[21,22,24,34] (Fig. 1g). Finally, and in agreement with the above results, exchange rates at TSSs positively correlate with active marks (H3K4me3), and anti-correlate with repressive histone marks (H3K27me3) and DNA methylation (Fig. 1h, i and Supplementary Fig. 2b, c).

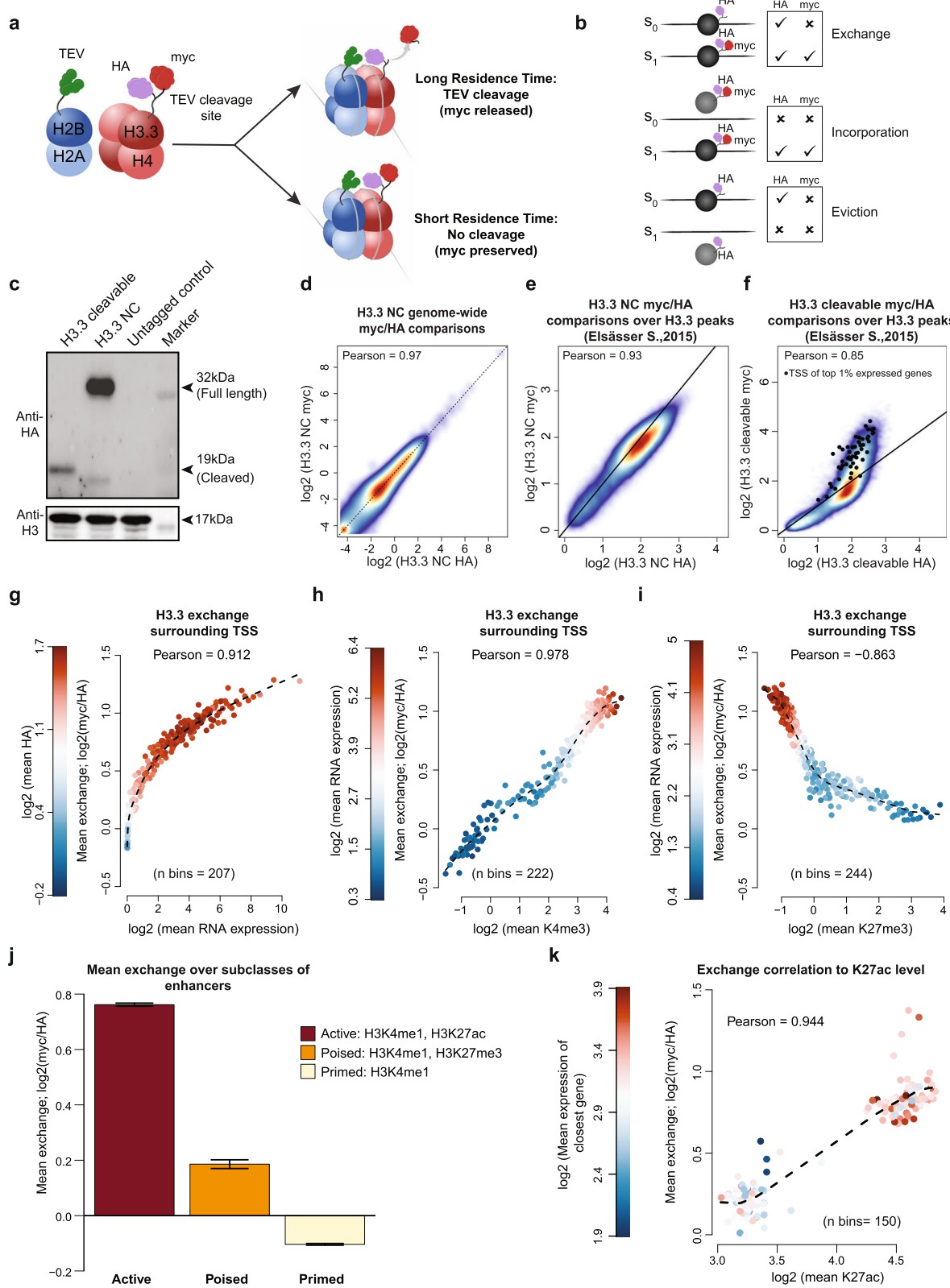

We next focused on distal enhancer elements, where H3.3 was also reported to be highly deposited[21,22,34,40–42]. In mESCs, enhancers exhibit various regulatory states defined by the combination of associated histone marks and are often classified as active, primed, or poised[43–47]. Active enhancers are typically co-marked by H3K27ac and H3K4me1, while enhancers primed for activation contain only H3K4me1-marked histones. Lastly, a subset of H3K4me1-marked histones termed

poised also contain Polycomb-derived H3K27me3 repressive marks and are not associated with active genes. Nevertheless, poised enhancers were shown to be bound by *trans*-activators and exhibit DNase-I hypersensitivity[44–46,48]. We observed the highest exchange within active enhancers, which positively correlated with increased expression of nearby genes (Fig. 1j, k). In contrast, primed enhancers showed overall less exchange of histone H3.3 (indicated by lower myc

**Fig. 1 | Sensor of histone exchange successfully reports on replacement levels of non-canonical H3.3 variant in mESCs. a** Schematic representation of the sensor system. **b** Schematic representation of histone exchange, integration and eviction as inferred by myc and HA signals. **c** Western blot analysis on TEV cleavage for H3.3 tags with cleavable and non-cleavable TEV cleavage site. Cleaved (20 kDa) and non-cleaved (35 kDa) version of histone H3.3 are shown. Anti-H3 was used as a loading control. 'Untagged control' are protein lysates from the isogenic untagged cell line. Blots are representative of at least three independent experiments. **d** Genome-wide comparison of myc and HA ChIP-seq signals in cells expressing non-cleavable (NC) sensor variant was performed by tiling genome into 5 kb windows ($n = 544480$). Mean read count is calculated for each window and represented as density plot. Shown is Pearson correlation calculated for all windows. Dashed is 1:1 reference line. **e, f** Relative myc and HA signal levels over annotated H3.3 peaks ($n = 71019$) for cells expressing NC (**e**) and cleavable sensor variant (**f**). Black dots represent H3.3 peaks overlapping with transcription start site (TSSs) of top 1% expressed genes. Represented is 1:1 reference line. **g** H3.3 exchange levels surrounding TSSs. Represented are genes binned by expression level ($n = 207$, 78-79 genes per bin). For each bin mean expression level is plotted against mean exchange (myc/HA, log2) and colored by mean HA signal. The dashed line is a trendline. Pearson correlation is calculated for all gene bins. **h, i** H3.3 exchange comparison to H3K4me3 (**h**) and H3K27me3 (**i**) histone marks within 2 kb windows centered at TSSs. Genes are binned by expression levels ($n = 222$ (**h**) and 244 (**i**)). Average H3K4me3 and H3K27me3 enrichment (log2, x-axis) is plotted against H3.3 exchange (y-axis) for each bin. Color denotes mean expression level of genes in each bin. The dashed line is a trendline. Pearson comparison used. **j** Histone exchange levels for active ($n = 12142$), primed ($n = 19723$), and poised ($n = 1015$) enhancers. Data are represented as mean ± SEM. **k** Bins of H3K27ac peaks overlapping annotated enhancers. Mean H3K27ac peak intensity level (log2) for each bin ($n = 150$) is then plotted against mean exchange and colored by mean RNA expression of the closest gene (log2). Represented in dashed is trendline. Source data are provided as a Source Data file.

to HA ratios), similar to background control regions (Fig. 1j and Supplementary Fig. 2d). Finally, although lacking active H3K27ac marks, poised enhancers exhibit lower but evident exchange levels, potentially reflecting the binding of chromatin remodelers or transcription factors at these sites. Taken together, the sensor system largely validated and further extended previous exchange measurements of histone H3.3 associated with enhancer and promoter elements.

## Variant-specific histone exchange patterns in open reading frames

We next set out to compare genome-wide exchange dynamics of canonical histone variants. Despite constituting the bulk of histones, genome-wide exchange of canonical variants has not been systematically charted. We therefore generated two additional sensor cell lines for the canonical histones H3.1 or H2B. For the former, the sensor was fused to histone H3.1 and the TEV-protease to H2B, and for the latter, the sensor was fused to histone H2B and the TEV-protease to histone H3.1, all as C-terminal fusions (Fig. 2a). Both systems were stably integrated into the *H11* locus, and proper expression and cleavage activity were validated at the protein level (Fig. 2b and Supplementary Fig. 3a). At the RNA level, the sensor and TEV enzyme showed at least 16- and 44-fold lower expression compared to the native histones, respectively (Supplementary Fig. 3b). Variable expression levels of sensor and TEV protease are unlikely to influence exchange measures as we observed that the vast majority of cleavable sensors exist in cleaved form, indicating that the sensor and protease have indeed been in proximity to each other (Fig. 1c and Fig. 2b). Furthermore, comprehensive read coverage was obtained for all analyzed samples, and controls containing non-cleavable myc showed the expected high correlations between myc and HA signals (Supplementary Fig. 3c, d). To identify exchange patterns associated with functional genomic regions for the three histone variants, we used a broad classification of regulatory elements, previously defined based on combinatorial patterns of histone marks[49] (Fig. 2c). All three variants show high exchange within active promoters and enhancer regions that are also enriched in H3.3 and H2B variants (Fig. 2c and Supplementary Fig. 3e). In heterochromatin regions (marked by H3K9me3), we detect high H3.1 exchange, whereas H3.3 and H2B histones are less exchanging. Interestingly, pronounced differences between histone variants were observed in regions associated with transcriptional elongation (marked by H3K36me3), where histone H2B exhibits rapid exchange while the other two variants showed little or no dynamics (Fig. 2c).

Focusing on the association between exchange and transcription, we compared the exchange levels in the vicinity of TSSs, clustered by gene expression. First, this analysis revealed that similar to histone H3.3 (Fig. 1g), exchange of both H3.1 and H2B scales with transcription levels (Fig. 2d). Second, this analysis confirmed variant-specific

exchange patterns in open reading frames. While H3.3 histones showed restricted exchange upstream of the TSSs, H3.1 exchange occurs adjacent to TSSs - both upstream and downstream- with low exchange within gene bodies. In contrast, histone H2B appears to exchange at similar levels at the TSS and in the gene-body. To further validate that differential signals between the sensors are not affected by cell line-specific differences in TEV localization, we profiled the distribution of HA-tagged H2B variant as a proxy for incorporation of tagged histones. This analysis confirmed a high correlation to native (untagged) H2B variant, indicating that the tagging of histones and expression under constitutive promoter did not influence their genome-wide distribution (Supplementary Fig. 4a). Moreover, H3.3 exchange surrounding TSSs does not correlate to H2B-HA signal, as opposed to the RNA expression levels, suggesting that the high myc levels cannot be attributed to lack of proximity of H2B-TEV variant (Supplementary Fig. 4b). Furthermore, when sorting genes by decreasing myc signal, we observed overall uniform distribution of H2B-HA, suggesting that the H3.3 and H2B tagged histones are indeed in proximity (Supplementary Fig. 4c). Similarly, sorting of genes by decreasing H3.1 or H2B myc signals also confirmed uniform distribution of complementary variants (Supplementary Fig. 4d,e), indicating that the high myc signals cannot be explained by non-uniform distribution of TEV-tagged nucleosomal subunit.

The results presented here are consistent with previous in vitro and in vivo findings in *S.cerevisiae*[36,39,50–53], demonstrating that RNA polymerase promotes the replacement of the more labile H2B histones in gene bodies, whereas H3 variants are mostly retained. While the functional significance of this observation remains to be elucidated, it is possible that retention of histone H3 within gene bodies might serve to stabilize an expression-promoting epigenetic landscape such as H3K36me3[28].

## Exchange of histone H3.1, but not H3.3 and H2B, at bivalent gene promoters is transcription-independent

Though most strongly associated with gene transcription, histone exchange was also proposed as a mechanism mediating gene silencing[24,25,39]. Therefore, we next sought to study histone exchange in association with gene repression. Polycomb protein groups (PcGs) – multicomponent protein collection that forms two complexes termed Polycomb repressive complex 1 (PRC1) and PRC2 and mediate gene repression through deposition of trimethylation of lysine 27 on histone H3 (H3K27me3) (reviewed in refs. 54, 55). At first, analysis of all gene promoters confirmed binding of PcGs components to be weakly associated with gene expression and exchange levels in the three variants (Fig. 2e). Amongst analyzed Polycomb proteins, SUZ12, EZH2, and JARID2 were shown to bind promoters of developmentally regulated genes, co-marked by both active H3K4me3 and repressive H3K27me3 histone modifications (termed bivalent promoters), and

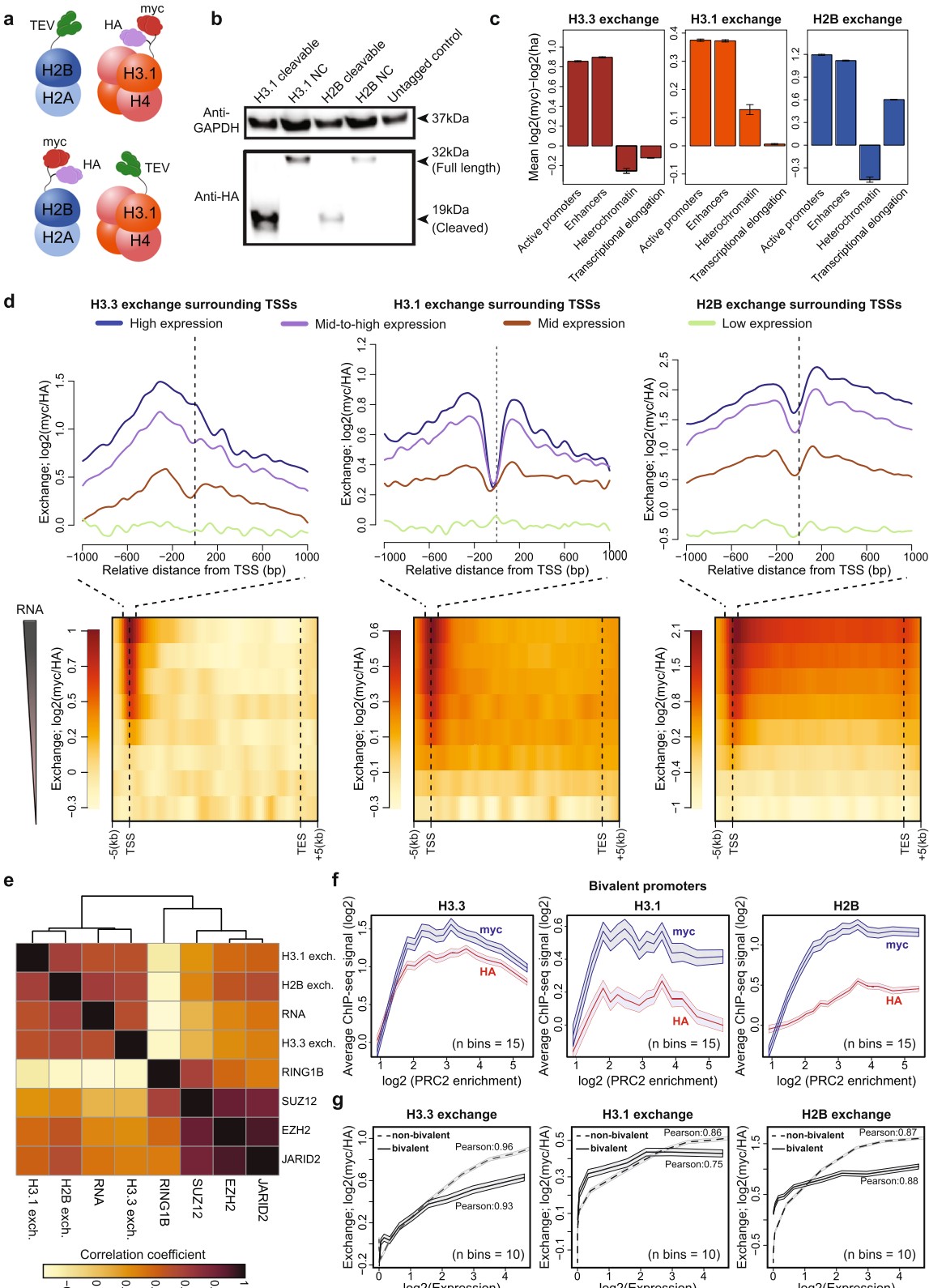

associated with H3.3 dynamics[24]. Given this observation, we focused the analysis on bivalent genes.

To study exchange at bivalent promoters, we first binned previously defined bivalent regions[56] by increased binding of SUZ12, EZH2, and JARID2 (Supplementary Fig. 4f,g), utilizing individual myc and HA signals to distinguish between additional modes of histone dynamics that included histone incorporation and eviction (Fig. 1b). Bins with

low PRC2 enrichment score (<2) showed elevation in both H3.3 HA and myc levels, demonstrating increasing H3.3 incorporation at these sites (Fig. 2f). Similarly, increasing HA and myc levels were also detected for H3.1 and H2B variants within bins with low PRC2 enrichment scores (<2 and 3.5, respectively). However, unlike H3.3, in these variants myc levels elevated more than HA, suggesting that incorporation is coupled with increased replacement at these sites. On the other hand, bins with

**Fig. 2 | Molecular tagging of core histone variants reveal differential exchange pattern of histone variants as a function of gene composition. a** Schematic representation of sensor system tagging H3.1 and H2B histone variants, respectively. **b** Western blot of whole-cell extracts showing cleavage of cleavable and NC cell lines for H3.1 and H2B tagged histone variants, respectively. Anti-GAPDH serves as a loading control. Control protein lysates from untagged isogenic cell lines were used. Blots are representative of at least three independent experiments. **c** H3.3, H3.1 and H2B barplot representation of mean exchange levels over different functional regions defined by histone modification marks from ref. 49. Number of regions are 22657 (Active promoters), 31340 (Enhancers), 2000 (Heterochromatin) and 32976 (Transcriptional elongation). For each type, exchange is normalized by subtracting mean exchange of analyzed regions to a mean exchange of control regions 20 kb downstream of the analyzed regions. Error bar denotes SEM. **d** Upper: Histone exchange around TSSs for four gene groups classified by their transcriptional level (number of genes per group from high to low: 5105, 5146, 4888, and 4453). Dashed lines indicate TSSs. Lower: Average histone exchange

levels for all genes ranked by expression level. Row corresponds to gene groups and columns to scaled-genic windows from 5 kb upstream to 5 kb downstream of each gene. **e** Pearson correlation heatmap between histone exchange ('Exch'; myc/HA, log2) and ChIP-seq counts of analyzed histone marks, RNA expression levels and Polycomb components across all gene promoters (defined as 1 kb region upstream of annotated TSS, $n = 19592$) ordered by hierarchical clustering. **f** Histone exchange at bivalent genes. Bivalent genes were binned (15 bins in total) and ranked by co-enrichment of SUZ12, EZH2 and JARID2 components of Polycomb complex 2 (See Methods). Mean myc and HA (log2) levels for each bin were then plotted over mean enrichment of PRC2 components. Shades denoting SEM. **g** Histone exchange at bivalent and non-bivalent genes. Promoters (1 kb upstream of TSS) of annotated bivalent and non-bivalent genes are grouped into 10 expression bins and mean exchange (myc/HA, log2) was calculated for each bin and plotted against mean RNA expression level of each bin. Shades denoting SEM. Source data are provided as a Source Data file.

mean PRC2 enrichment (from 2 to 3.5 for H3.3 and H3.1 variants and >3.5 for H2B variant) show constant histone occupancy and largely invariant myc-to-HA ratio, demonstrating that after a certain level, increased binding of PRC2 does not lead to higher exchange. Finally, bins with high PRC2 binding showed decreased HA signal for the H3.3 and H3.1 variants. As myc level of H3.3 is also decreasing this suggests reduced incorporation of H3.3 at these sites. Contrary, myc levels of H3.1 remained mostly invariant, suggesting increased eviction of this variant or of tagged histones (Fig. 2f), perhaps due to direct competition between histones and PRC2 for DNA binding.

We next binned promoters of bivalent genes by their expression levels and compared their associated exchange levels to those of non-bivalent genes with a similar expression. This analysis showed that exchange levels of H3.3 and H2B variants at bivalent genes increased with transcription, but to a lesser extent compared to the non-bivalent genes. Contrary, exchange level of H3.1 remained constant and largely independent of increased transcription (Fig. 2g). Together, our data is most consistent with H3.3 and H2B exchange at bivalent genes correlating with increased transcription rather than with PRC2 binding, further extending previous analysis of chromatin properties at bivalent genes.

## Asymmetric exchange surrounding CTCF binding sites, irrespective of high-order chromatin organization

In eukaryotes, gene activity is largely influenced by its spatial positioning and genomic architecture that are controlled by structural regulatory elements of intergenic regions[57–59]. To understand how structural features of chromatin relate to histones exchange, we focused on the CCCTC-binding factor (CTCF) that is considered a core architectural protein, previously shown to introduce a prototypic, regularly spaced nucleosome organization[60–66]. Indeed, plotting HA signal surrounding CTCF motif occurrences recapitulated similar symmetrical patterns in the three histone variants (Fig. 3a and Supplementary Fig. 5a). We next calculated histone exchange levels within 2 kb windows centered on the CTCF binding sites, while filtering out motifs that reside in genic regions, and calculated exchange for the three variants. We detected low yet significant exchange of H3.3 and H3.1 variants and high exchange of H2B variant, comparing each to control regions shifted 20 kb from the CTCF sites (Fig. 3b). CTCF plays a key role in mediating chromatin folding into isolated domains of preferential long-range interactions, known as topologically associated domains (TADs)[60,61,63,64,67–71]. In the prevailing model, CTCF proteins create a base of chromatin loops by thwarting the progression of cohesin-mediated loop extrusion[60,63,67,72–74]. Importantly, CTCF binding motifs mediating loop extrusion display convergent orientation (facing inwards) that directs the binding polarity of CTCF[64,73,75] (Fig. 3c).

Analyzing myc and HA signals associated with nucleosomes surrounding CTCF sites, while accounting for sequence orientation (see Methods), we found that the exchange is mainly restricted up to −2 to +2 nucleosomes. Interestingly, comparing downstream and upstream nucleosomes with respect to the CTCF binding motif revealed significant asymmetric histone exchange patterns associated with all three histone variants ($p$-value of two tailed $T$-test < 0.05, Fig. 3d). To test whether the observed exchange could be related to CTCF binding and not the underlying sequence, we measured exchange levels within CTCF motifs that are differentially bound by CTCF (see Methods). This analysis showed high histone exchange associated with CTCF bound motifs, whereby non-bound motifs are depleted of exchange (Fig. 3e). Next, for each variant, we selected the top 10% of sequences harboring asymmetric H3.3 exchange at the +1 and +2 positions and calculated the relative enrichment of various histone modifications around these sequences, as compared to bottom 10% of asymmetrically exchanged sites. This analysis identified increased exchange associated with high levels of open chromatin modifications (H3K4me3 and H3K27ac) but not with transcription (H3K79me2 and H3K36me3) or repressive marks (H3K27me3) (Fig. 3f and Supplementary Fig. 5b). We note that since annotated genes are excluded from this analysis, the observed asymmetric exchange may be related to active enhancer elements residing adjacent to TAD borders, as previously shown[61].

At a larger scale, TADs are organized into two chromatin compartments termed compartment A and compartment B. Compartment A is associated with open chromatin, and compartment B is defined by lamina-associated domains of late replicating heterochromatin[67,68,71,76]. To assess histone exchange in the vicinity of CTCF binding sites, and whether it differs between open and closed chromatin, we analyzed annotated CTCF sites accounting for A and B compartments (see Methods). Interestingly, this analysis showed comparable exchange in both chromatin compartments for H3.3 and H3.1, whereas H2B exchanges more in compartment B (Fig. 3g). Together, our analysis identifies histone exchange surrounding CTCF-bound motifs in non-transcribed intergenic regions, irrespective of open or closed chromatin. CTCF binding and chromatin remodeling factors promote chromatin folding, but the direct effect on histone dynamics is less clear[77–79]. For example, SNF2H chromatin remodeler was shown to be critical for nucleosome organization and influence binding of CTCF[77]. Therefore, the observed histone exchange in the vicinity of CTCF sites could be mediated by additional *trans*-acting factors such as SNF2H[77]. Perturbation studies involving CTCF and other putative *trans* factors in cells harboring histone exchange sensors could provide valuable insight into this regulation.

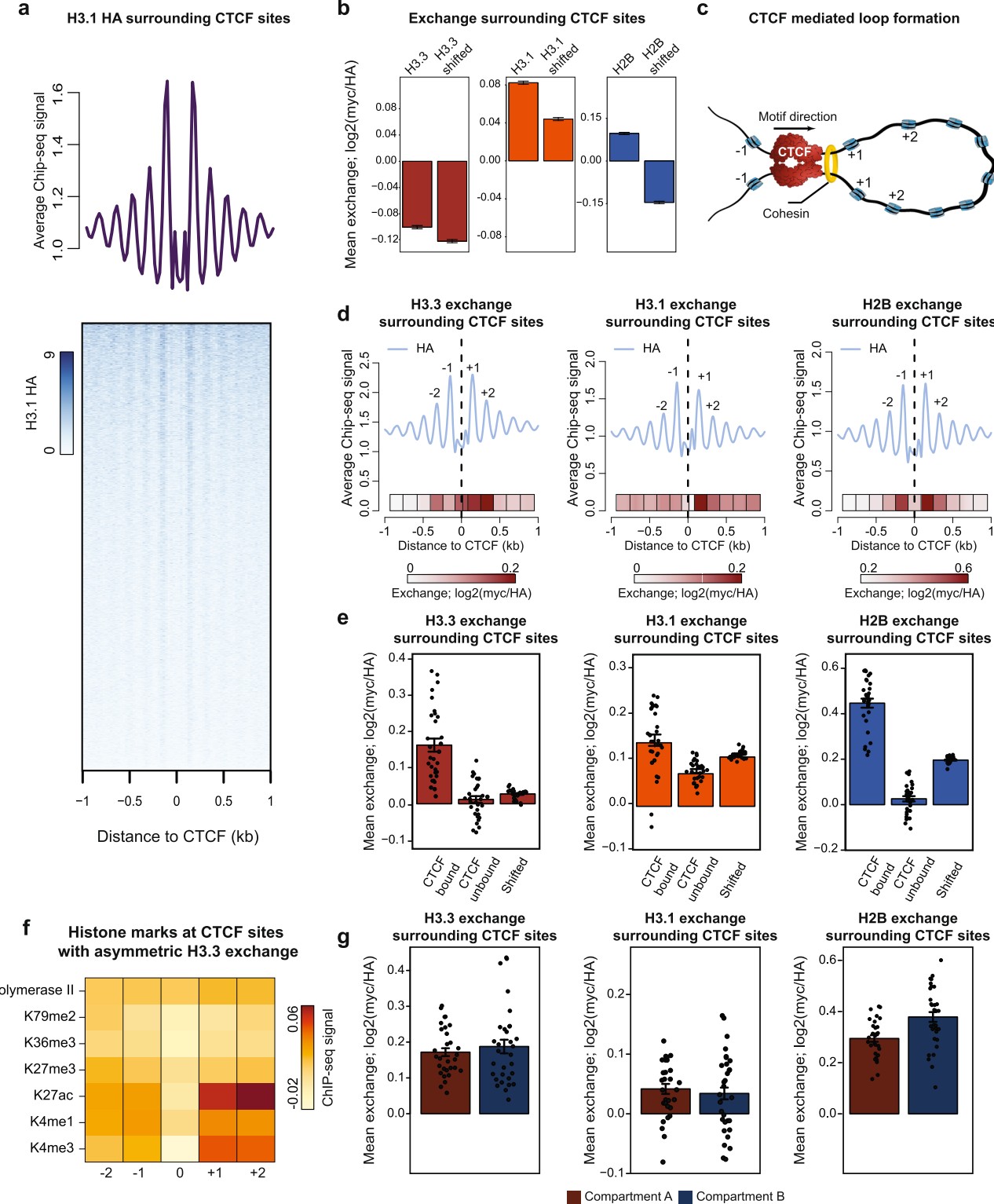

**a** H3.1 HA surrounding CTCF sites

**b** Exchange surrounding CTCF sites

**c** CTCF mediated loop formation

**d** H3.3 exchange surrounding CTCF sites / H3.1 exchange surrounding CTCF sites / H2B exchange surrounding CTCF sites

**e** H3.3 exchange surrounding CTCF sites / H3.1 exchange surrounding CTCF sites / H2B exchange surrounding CTCF sites

**f** Histone marks at CTCF sites with asymmetric H3.3 exchange

**g** H3.3 exchange surrounding CTCF sites / H3.1 exchange surrounding CTCF sites / H2B exchange surrounding CTCF sites

Compartment A  Compartment B

## High H3.1 and H2B exchange within heterochromatin is associated with H3.3 occupancy

Heterochromatin is frequently associated with compacted nucleosomes with low exchange rates[80]. However, consistent with our findings of high H3.1 exchange within H3K9me3 regions, as well as H3.1 and H2B exchange at compartment B CTCF binding sites (Figs. 2c and 3g), recent work identified histone dynamics within heterochromatic regions associated with the silencing of transposable elements[31,81]. It was suggested that chromatin remodelers enforce the eviction of canonical histone variants and incorporation of newly synthesized H3.3, ultimately rendering DNA in a closed state[81]. We examined this in our system by analyzing H3.3 occupancy (as measured by HA signal) at DNA repeats. To this end, we mapped reads to a comprehensive database of murine repetitive sequences[82] and compared them to previously reported ChIP-seq data of H3.3[31]. We observed an overall high correlation between the two datasets ($\rho = 0.79$) and in agreement with previous observations[22,31,83–86], detected telomeric repeats and sub-families of intracisternal A-type

**Fig. 3 | Asymmetric exchange surrounding CTCF binding sites. a** H3.1 histone occupancy within 2 kb window of annotated CTCF binding regions (number of all sites 83810). Average metaprofile for all the sites (upper panel) and heatmap counts of HA over individual sites (lower panel) are shown. Rows of heatmap are sorted by total H3.1 HA counts in a decreasing manner. Limits of H3.1 HA signals are set to 9. **b** Exchange levels of histone variants within 2 kb window centered at CTCF sites outside genetic regions and control region shifted 20 kb downstream of annotated CTCF sites (number of CTCF sites outside genetic regions 53917). Data are represented as mean ± SEM. **c** Schematic representation of CTCF-mediated chromatin loop: Translocation of cohesin on DNA forms nascent chromatin loop until blocked by a pair of CTCF proteins, bound in a convergent orientation. Nucleosomes downstream (+1, +2) and upstream (−1) of CTCF binding sites are shown. **d** Histone occupancy (metaprofile, upper) and exchange (heatmap, lower) at 2 kb window of CTCF binding regions within non-transcribed intergenic regions ($n = 53917$). Plots are corrected for motif orientation of CTCF binding sites. Exchange (myc/HA, log2)

was calculated for each nucleosome position and represented as heatmap. **e** Mean histone exchange (myc/HA, log2) calculated for four different nucleosomes surrounding CTCF-occupied and CTCF-unoccupied binding motifs (see Methods for details). Exchange was calculated for 31 bins covering the four nucleosomes. Bars indicate mean values for these 31 bins and error bars represent SEM. **f** Histone modification profile within DNA loops that show asymmetric H3.3 exchange at +1 and +2 sites. Enrichment (IP/input, log2) of indicated histone marks and RNA Pol II of top 10% H3.3 asymmetrically exchanged CTCF sites are compared to bottom 10% H3.3 asymmetrically exchanged CTCF sites. Fold change surrounding +1, +2, −1, and −2 nucleosome sites are shown. **g** Histone exchange at CTCF sites overlapping with A-compartment ($n = 32606$) or B-compartment ($n = 11976$) as defined in ref. [67]. Mean exchange (myc/HA, log2) was calculated for 31 bins covering four nucleosomes and further normalized by subtracting to mean exchange (myc/HA, log2) within control regions shifted 20 kb downstream. Error bar denotes SEM. Source data are provided as a Source Data file.

---

particles (IAPs) and RLTRs to exhibit enriched levels of H3.3 occupancy (Fig. 4a).

Focusing next on exchange levels, we found a negative correlation between occupancy and exchange for histone H3.3 (Fig. 4b). Conversely, the exchange of canonical H3.1 and H2B variants positively correlates with H3.3 occupancy (Fig. 4c,d). This was especially evident for IAP sub-families, which exhibited the highest exchange levels of histone H3.1, but low exchange of H3.3 (Supplementary Fig. 6a). Profiling the landscape of histone occupancy surrounding IAPs showed enriched incorporation of H3.3 at the 5' and 3' boundaries of the elements (Fig. 4e). To examine whether exchange profiles of H3.1 and H2B coincide with the polar H3.3 distribution, we aligned reads over a curated set of 2638 full-length IAP elements[81] (see Methods) and found exchange of H3.1 and H2B to peak at the 5', but not the 3' border region (Fig. 4e).

We extended our analysis, focusing on regions flanking individual repeats that could be unambiguously mapped with our short sequencing reads. IAP sequences were ordered by H3.3 levels from ref. [31], demonstrating the inverse relation to H3.3 and an overall positive correlation with H3.1 exchange levels (Fig. 4f). Sorting by individual sequence also uncovered that H2B exchange is restricted to the subset of elements that exhibit the highest H3.3 occupancy (Fig. 4f). Finally, high H3.3 signal at IAPs also correlated with H3K9me3 marks and low DNA accessibility, in line with their heterochromatic state (Fig. 4g). Together, our data uncover surprisingly high levels of H3.1 and H2B exchange within heterochromatin and repeat elements, which are associated with the incorporation of non-canonical histone H3.3. It remains unclear how this exchange of canonical variants is mechanistically associated with heterochromatin and silencing of transposable elements.

### Knockout of the H3.3 chaperone HIRA results in decreased H3.1 dynamics in open chromatin

To ask whether the relationship between incorporation of histone H3.3 and exchange of canonical variants is restricted to repeat sequences and H3K9me3 domains, we focused on active and primed enhancers and promoter elements, where H3.3 is highly abundant[21,22,24,26,34,40–42,87–90] (Supplementary Fig. 3e). Ordering sequences by increased levels of H3.3 occupancy (as measured by HA) identified a positive correlation with both occupancy and exchange of histones H3.1 and H2B (Fig. 5a). But interestingly, such a trend was not observed when ordering the same regions by increased H3.1 and H2B occupancy levels (Supplementary Fig. 7a,b). We hypothesized that deposition of H3.3 at enhancers and promoters could promote increased histone dynamics at these regions. To further investigate this potential link, we considered HIRA - the chaperone primarily responsible for incorporating histone H3.3 in open chromatin regions[22–24,28,41,87–89].

H3.3 was highly enriched in our dataset at previously annotated HIRA bound regions compared to H3.1 or H2B, validating specific

interaction of HIRA with the H3.3 variant (Fig. 5b). To study association between exchange and H3.3 deposition at open chromatin, we performed CRISPR-Cas9 mediated HIRA knockout in cell lines comprising either the H3.3 or H3.1 sensor systems (Fig. 5c). Following validation of HIRA depletion at the DNA and protein levels (Fig. 5d and Supplementary Fig. 7c,d), ChIP sequencing was performed on two independent knockout clones from each sensor cell line. As expected, we observed a dramatic reduction of H3.3 occupancy at enhancers and promoter regions in HIRA mutant cells. Notably, no significant change in occupancy was observed within H3K9me3 peaks, supporting the functional separation between HIRA and DAXX/ATRX chaperones – the latter strongly implicated in incorporating H3.3 into heterochromatin regions[22,84,85,91] (Fig. 5e). In agreement with previous findings[24], we found increased H3.1 occupancy at enhancers and promoters (but not H3K9me3 regions) in HIRA mutant cells, implying a mechanism compensating for the absence of H3.3 (Fig. 5f).

We next assessed the effect of HIRA depletion on histone exchange. Consistent with measured occupancy patterns, we found a marked decrease in H3.3 dynamics within active sequences but not in heterochromatic regions (Fig. 5g). But interestingly, despite the increased occupancy in active and primed enhancers and promoters, we found a significant reduction in the exchange levels of histone H3.1 in these regions (Fig. 5h). A working model emerging from these results suggests that HIRA-mediated incorporation of histone H3.3 in open chromatin promotes the eviction of canonical H3 histones. This is consistent with our above findings that H3.3 deposition enhances histone dynamics (Figs. 4e, f, and 5a).

### H3.3 but not H3.1 shows predominant exchange in non-dividing hepatocytes in vivo

Due to well-appreciated drawbacks of commonly used methods to profile histone dynamics in vivo, most of our current understanding of this key process in mammals is derived from cell lines. To test whether the sensor system allows robust profiling of histone exchange in vivo, we generated transgenic mice carrying either H3.3 or H3.1 sensors. Transgenic mice were fertile and overtly healthy, and modified alleles were transmitted at normal Mendelian ratios, suggesting little or no toxicity. Previous metabolic labeling in post-mitotic mammalian cells, revealed canonical histone variants as long-lived proteins with low turnover[92–94]. Given the intrinsic properties of cell-cycle dependent and independent expression for the two H3 variants, we speculated that the exchange of replication-dependent H3.1 variant will be restricted to dividing cells, whereby H3.3 would be exchanged independently of cell division. To test this, we first isolated post mitotic hepatocytes cells from adult mice and profiled turnover for the two variants (Fig. 6a). As expected, we found correlation between H3.3 exchange and gene transcription which also positively correlated to H3K4me3 active histone marks (Fig. 6b and Supplementary Fig. 8a). Additionally, we found high exchange within H3K27ac peaks, marking

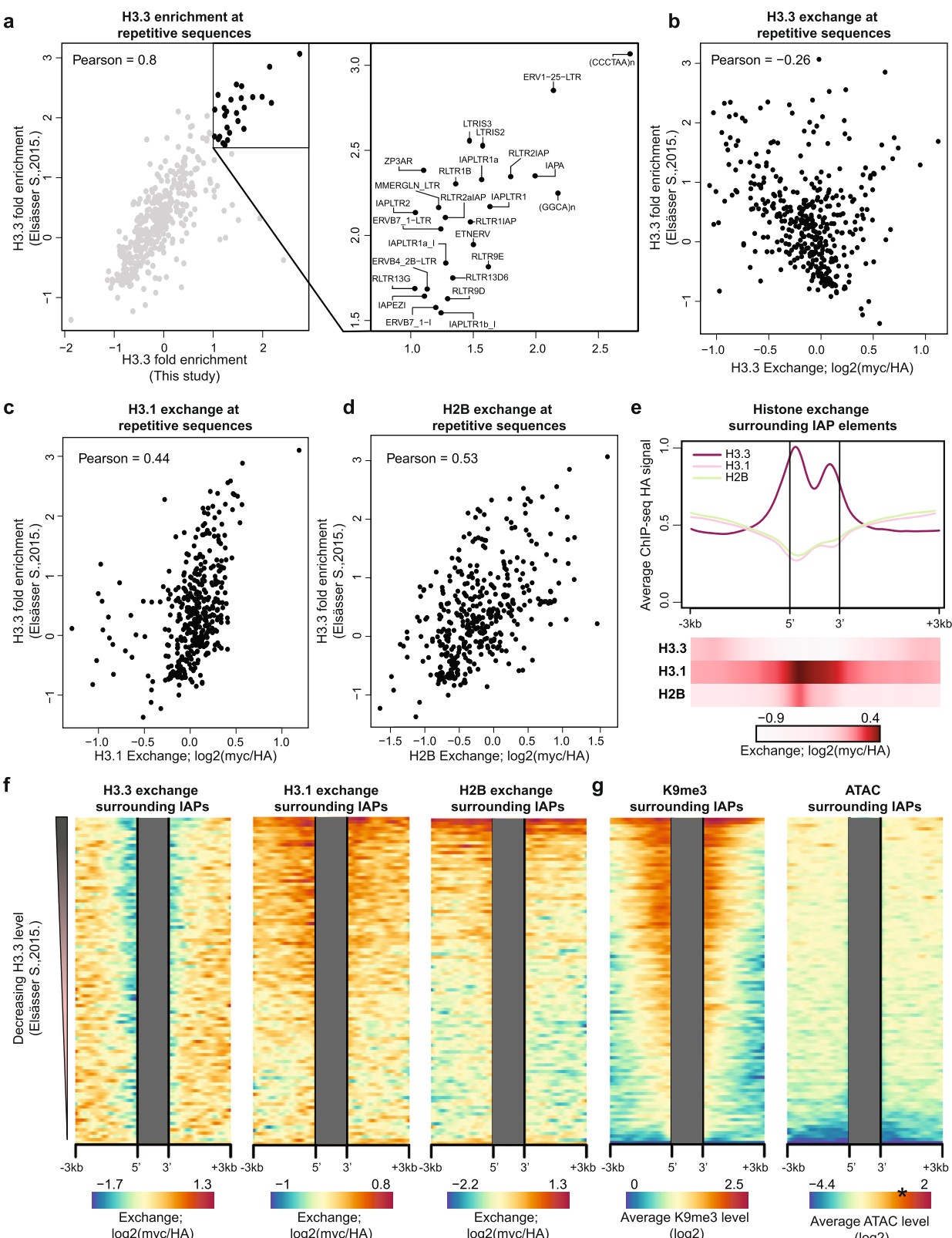

regulatory regions (Supplementary Fig. 8b). Contrary, no correlation was found between H3.1 exchange and gene transcription or regulatory H3K27ac-marked sites (Fig. 6c and Supplementary Fig. 8a,b). This is in line with previous results, suggesting low turnover of H3.1 in hepatocytes[93–95]. Importantly, despite relatively low myc levels, HA signal was readily detected within profiled genomic regions, indicating appropriate deposition of H3.1 at these sites, probably while the cells

were still dividing (Supplementary Fig. 8a). We next isolated dividing mouse embryonic fibroblasts from E12.5 embryos harboring H3.1 sensors (Fig. 6d). The analysis of H3.1 exchange in dividing cells indeed confirmed correlation between H3.1 exchange, gene transcription and active histone marks (Fig. 6e and Supplementary Fig. 8c). These results demonstrate the utility of our system for broad in vivo applications in mice. Furthermore, we validated that the incorporation

**Fig. 4 | Increased exchange of canonical histone variants in H3.3 occupied repetitive sequences of heterochromatin. a** Correlation plot of H3.3 enrichment over annotated mouse repetitive sequences ($n = 405$) in two datasets. For Elsässer dataset enrichment was calculated as log2 fold change over input and for this study enrichment was calculated as log2 difference between H3.3 HA and H3.1 HA read counts. Zoomed in are repetitive sequences with the highest enrichment in both datasets. Pearson correlation coefficient is indicated. **b**–**d** Correlation between H3.3 fold enrichment and H3.3 (**b**), H3.1 (**c**) or H2B (**d**) exchange (myc/HA, log2) within annotated mouse repetitive sequences ($n = 405$). Pearson correlation coefficients are indicated. **e** H3.3, H3.1 and H2B occupancy (metaprofile, upper) and histone exchange levels (heatmap, lower) over annotated IAP elements from ref. 81 ($n = 2638$) and surrounding regions (See Methods). **f** Histone exchange (myc/HA,

log2) within uniquely mapped regions surrounding IAP elements. Each row represent collection of IAP elements that are ordered and binned based on H3.3 counts from ref. 31 (total of 100 bins covering 2638 IAPs). For each IAP bin, average exchange levels were calculated. Gray boxes mark internal IAP regions of repetitive sequences without unique alignment. **g** Enrichment of H3K9me3 and ATAC signal at DNA regions flanking IAP repeats ordered and binned as in **f**. Only uniquely mapped reads were used. Color gradient of ATAC-seq dataset was rescaled to match ATAC-seq signal density over 2 kb enhancers window ordered and binned in the same manner as IAP repeats. * represents median ATAC-seq signal over binned enhancer elements. Of note 99.83% of IAP bins are less accessible than enhancer regions. Source data are provided as a Source Data file.

and exchange of modified H3.1 variant is restricted to the dividing cells whereby H3.3 is exchanging independently of replication status, as expected.

## Discussion

Current mapping of the exchange landscape of histone variants is largely incomplete. In this study, we used sensors that continuously report on histone occupancy (measured by HA tag), incorporation (myc tag), and exchange (myc/HA ratio), at each genomic locus, providing a comprehensive genome-wide view of nucleosome occupancy and dynamics. Applying the sensors to H3.3, H3.1, and H2B histone variants, identified both transcription-associated and transcription-independent exchange. Though transcription-associated exchange was expected, we found variation in the exchange patterns between the three variants: H3.3 exchanges exclusively in the promoters, H3.1 in the vicinity of TSSs, and H2B equally in promoters and open reading frames. These results are consistent with previous findings[36,39,50–53], demonstrating that transcription and RNA polymerase promote the replacement of the more labile H2B in gene bodies, whereas H3 variants are mostly retained. Nucleosome remodeling was previously shown to facilitate transcriptional elongation[51,96]. However, whilst disruption of nucleosomes is necessary for passage of RNA polymerase within gene bodies, it opposes chromatin integrity and retention of histone marks. We therefore speculate that the restricted eviction of H2A-H2B dimers serves to loosen the DNA while retaining modification-enriched H3-H4 subunits in place.

Histones are also exchanging within regulatory elements in a transcription-independent manner, albeit at lower levels, as in the case of H3.3 exchange at poised enhancers and H3.1 exchange at bivalent promoters of lowly expressed genes. Exchange at these regions could result from competition between repressors and histones for binding to the same regions, resulting in decreased histone occupancy as exemplified for H3.1 and H3.3 variants within bivalent promoters of high PRC2 binding. Since in mESCs poised enhancers and Polycomb regulated promoters are implicated in cell differentiation potency[45,46,97], exchange within these sites potentially poises local chromatin for binding of lineage-specific transcription factors (TFs). Yet, perturbation studies on factors regulating exchange at these sites and their effect on differentiation are needed to test this hypothesis.

Despite the high exchange levels of histone H3.3 in active genes and distal regulatory elements, depletion of H3.3 was shown to introduce little effect on the overall gene expression program in mESCs[24,40]. Rather than maintaining an already established transcriptional program, H3.3 seems to promote gene activation de novo[90,98–100]. This is also supported by recent data in which HIRA-mediated depletion of H3.3 was associated with small, yet significant, loss of accessibility and TF binding at gene promoters and altered cellular differentiation[101]. Here, depleting HIRA in mESCs resulted in a significant reduction in histone exchange at open chromatin, suggesting that H3.3 deposition promotes nucleosome dynamics at these sites. Given the role of H3.3 in transcriptional activation, we propose that H3.3-promoted exchange at regulatory elements primes DNA accessibility for binding of TFs.

Hence, exchange in these sites is probably occurring prior to, and not as a consequence of, TF binding.

Intriguingly, we detected exchange dynamics for H3.1 and H2B variants in repetitive sequences embedded within lamina-associated, H3K9me3 marked heterochromatin regions. Silencing of repeat sequences takes place in a dynamic chromatin environment in which nucleosomes are disrupted and re-assembled by complex protein machinery involving chromatin remodelers, TFs, and histone chaperones[81,102,103]. We demonstrate that such chromatin resetting is associated with continuous exchange of canonical H3.1 and H2B histones but not exchanging H3.3 histone. The interdependency between canonical and non-canonical variants in this context is less clear, but it is likely that replacement of core histone variants creates DNA-accessible regions that promote binding of H3.3 and its associated silencing complex involving ATRX/DAXX chaperones and methyl-transferases. In the case of IAP elements, DAXX appears to play a major role in silencing and involves interaction with H3.3, whose function is to stabilize DAXX protein levels[91]. In this context, the effect of H3.3 on IAP silencing is rather indirect and does not require binding to DNA[91]. This argues against functional importance of low exchange levels for H3.3 within IAPs, as its relatively long residence time may reflect stable positioning of the DAXX-associated repressive complex.

Finally, we showed that the sensor system can be used successfully in mice. By constructing two transgenic mice models for canonical H3.1 and non-canonical H3.3 variants, we showed that exchange of H3.1 is restricted to dividing cells, whereby H3.3 was also exchanging in non-dividing hepatocytes. The in vivo results are complementing previous studies on histone dynamics, demonstrating that the H3.3 but not H3.1 variant is exchanging in post-mitotic tissues[32,92–95]. The presented sensor system requires only a single sample, largely precluding temporal delays in measuring exchange levels. This facilitates the comprehensive study of dynamic processes, thus providing exciting opportunities for future applications in development and adult tissues, in which histone exchange has thus far remained largely unattainable.

## Methods

### Ethics oversight

All research in this study complies with all relevant ethical regulations. All biohazard protocols were approved by the Weizmann Institute of Science. All animal procedures were approved by the relevant Weizmann Institute IACUC (#08241020-2 and #02610320-2).

### Culture of mESCs

Mouse ES cells were cultured on irradiated mouse embryonic fibroblasts (MEFs) under standard conditions: 500 ml DMEM (Gibco, 41965-039), 20% fetal bovine serum (Biological Industries, 04-001-1 A), 10 mg recombinant leukemia inhibitory factor (LIF, homemade), 0.1 mM beta-mercaptoethanol (Gibco, 31350-010), penicillin/streptomycin (Biological Industries, 03-031-1B), 1 mM L-glutamine (Biological Industries, 03-020-1B), and 1% nonessential amino acids (Biological Industries, 01-340-1B).

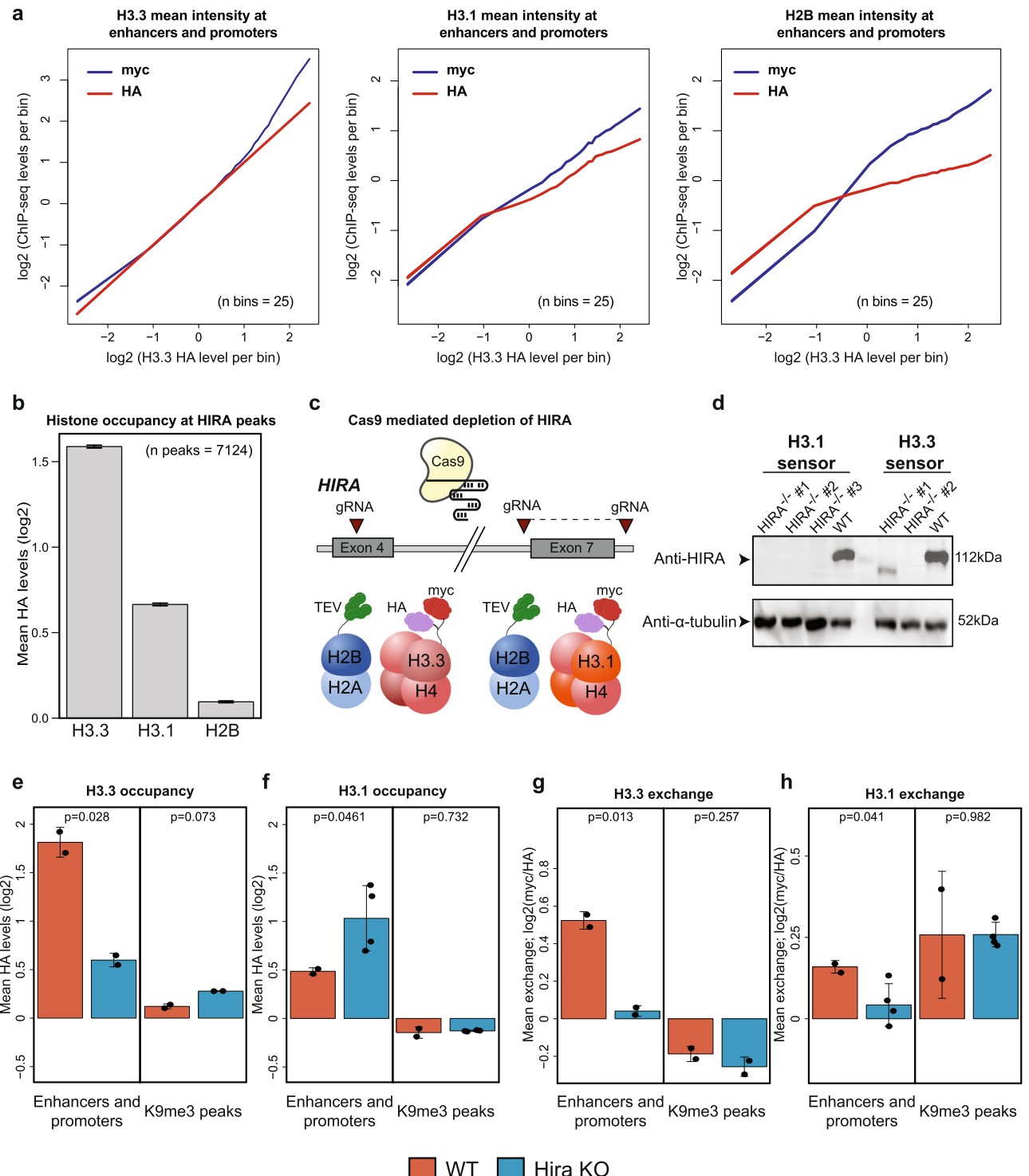

**Fig. 5 | HIRA-mediated deposition of H3.3 variant at regulatory elements is coupled with replacement of H3.1.** **a** H3.3, H3.1 and H2B occupancy and exchange over H3.3 occupied enhancers and promoters (*n* = 51457). Enhancer and promoter sequences were binned (25 in total) by H3.3 HA level (log2) and average myc and HA signals (log2) of each histone variant were calculated for each bin. Shaded area denotes SEM. **b** Mean occupancy of H3.3, H3.1 and H2B (measured by HA, log2) over 2 kb regions centered at HIRA peaks (*n* = 7124). Error bar denotes SEM. **c** Schematic representation of CRISPR/Cas9 strategy for HIRA depletion. **d** Western blot of whole-cell extract for control and Hira⁻/⁻ ESCs. Anti-α-tubulin was used as a loading control. Blots were performed once. **e**–**h** Histone occupancy (**e**, **f**) and exchange

(**g**, **h**) for H3.3 and H3.1 in WT and Hira⁻/⁻ mESCs at indicated genomic regions. Mean values of HA and exchange are normalized by subtracting mean values of appropriate control regions (see Methods). Dots representing mean HA or exchange levels across the indicated regions for each individual biological repeat (number of H3.1-cleavable and H3.3-cleavable WT repeats = 2; number of H3.3-cleavable HIRA KO repeats = 2; number of H3.1-cleavable HIRA KO repeats = 4). Error bars denote SD. Indicated are *p*-values of two-sided *T*-test. Number of regions analyzed are 15305 (H3.3 overlapping enhancers and promoters), 48492 (H3.3 non-overlapping enhancers and promoters), 18226 (H3.3-H3K9me3 overlapped region) and 53275 (H3K9me3-only region). Source data are provided as a Source Data file.

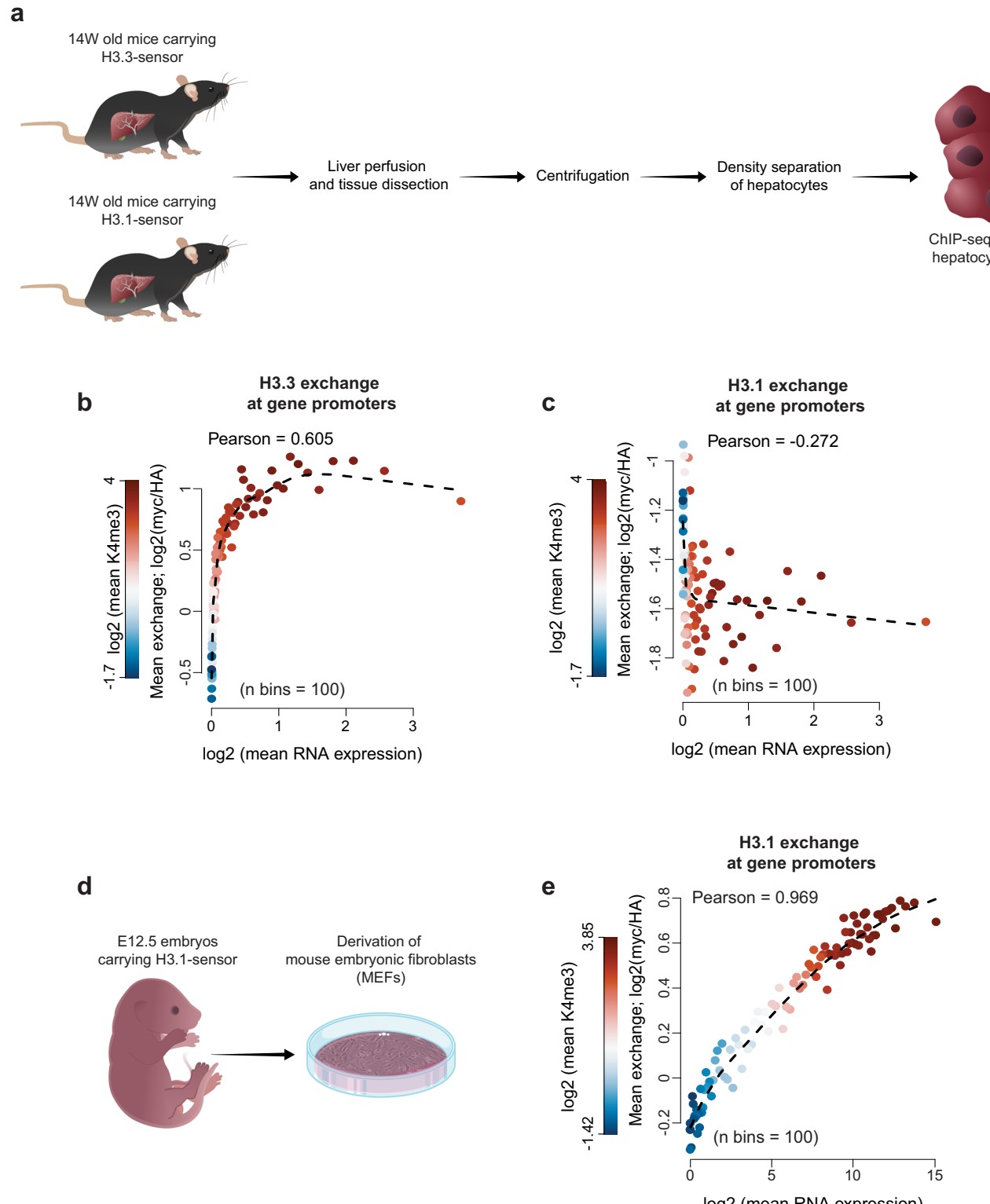

**Fig. 6 | In vivo implementation of the sensor system. a** Schematic representation of isolation of hepatocytes from mice carrying the H3.1-tagged or H3.3-tagged exchange sensor. Perfused liver of 14 week old mice was mechanically dissected and trypsinized to promote dissociation of liver cells. Dissociated cells were filtered and spun down to enrich for hepatocytes, and mixed with Percoll solution for density separation. After centrifugation, the pellet containing pure hepatocytes was saved for ChIP analysis. **b**, **c** H3.3 (**b**) and H3.1 (**c**) exchange levels for 2 kb windows surrounding TSSs of hepatocytes. Represented are gene collections grouped by expression level in hepatocytes (*n* = 100, 180-181 genes per group). For each gene collection, the mean expression level was plotted against mean

exchange (myc/HA, log2) and colored by mean H3K4me3 active histone mark (log2). The dashed line is a trendline. Pearson correlation is calculated for all gene bins. **d** Schematic representation of MEF derivation from E12.5 embryos carrying the H3.1-tagged sensor. **e** H3.1 exchange levels for 2 kb windows surrounding TSSs of dividing MEFs. Genes are binned by expression levels in MEFs (*n* = 100, 223-224 genes per group) and for each bin mean exchange (myc/HA, log2) was plotted against mean RNA expression level and colored by mean H3K4me3 level (log2). The dashed line is a trendline. Pearson correlation is calculated for all gene bins. Source data are provided as a Source Data file.

## Cloning of plasmids

Donor vectors containing H3.3 variant tagged with TEV-cleavable marker and H2B-TEV variants were cloned as follows: EF1-a driven H2Bb-ASGGSGGGS-TEV-PolyA fragment was synthetized as synthetic gBlocks Gene Fragment (IDT) and cloned into an in-house donor vector containing two attB sites. Coding sequence of H3.3B histone variant was PCR amplified from cDNA with primers PRIMER1 and PRIMER2. 3xHA-TEVcleavageSite-8xmyc was PCR amplified from synthetic gBlocks Gene Fragments (IDT) with primers PRIMER3 and PRIMER4. Separate gBlocks were used for cleavable (TENLYFQSGTRRW) and NC (TGGSGGGSGTRRW) variants. All sensor and TEV sequences are identical to those reported in ref. 39 and can be found therein. Backbone vector containing EF1a-H2B-TEV-PolyA was linearized and used for insertion of H3.3B and 3xHA-TEVcleavageSite-8xmyc fragments in three-piece ligation cloning. Finally, PGK-PURO was digested from #70148, Addgene plasmid and cloned into EF1a-H2B-TEV-PolyA-CAG-H3.3B-3xHA-TEVcleavageSite-8xmyc-PolyA vector.

To create H3.1 donor vector, EF1a-H2B-TEV-PolyA-CAG plasmid was used as a backbone. First, Hist1h3g sequence variant of H3.1 PCR was amplified from genomic DNA with PRIMER5 and PRIMER6. 3xHA-TEVcleavageSite-8xmyc was PCR amplified from gBlocks Gene Fragments with primers PRIMER3 and PRIMER4. Three-piece ligation was then used to clone these fragments.

When switching between H3.1 and H2B two-step cloning was performed. First, H3.1-3xHA-TEVcleavageSite-8xmyc was cut out from EF1a-H2B-TEV-PolyA-CAG-H3.1-3xHA-TEVcleavageSite-8xmyc-PolyA and served as a backbone vector. H2B was amplified from EF1a-H2B-TEV-PolyA-CAG-H3.3B-3xHA-TEVcleavageSite-8xmyc-PolyA plasmid with PRIMER7 and PRIMER8. 3xHA-TEVcleavageSite-8xmyc was then PCR amplified from corresponding gBlocks (either cleavable or NC) with PRIMER9 and PRIMER4. H2B and 3xHA-TEVcleavageSite-8xmyc fragments were then used in three-fragment ligations with the backbone vector. In the second cloning step, TEV-coupled H2B was excised and replaced with H3.1, previously PCR amplified with PRIMER10 and PRIMER11. All plasmids were Sanger sequenced before integration into mESCs.

## Generation of knock-in mESCs

Cell lines expressing the sensor system were generated by targeting into *Hippo11* safe harbor locus by site-specific recombination involving attB and attP attachment sites. We utilized previously generated V6.5 cell line (originally from Jaenisch lab, MIT. RRID:CVCL_C865) carrying three partial attP sites within H11 region to allow efficient site-specific recombination with two attB sites at donor plasmid (Supplementary Figs. 1a and 3a) as described in ref. 104. Donor plasmids containing genes of interests were co-transfected with φC31-bacteriophage derived integrase in *H11* V6.5 mESCs using TransIT-X2 Transfection Reagent (Mirus Bio, MIR6003) according to the provider's instructions. Twenty-four hours post-transfection, standard mESCs medium was supplemented with 1:1000 of 1 m/ml of Puromycin to select for integration.

## Western blot

For extraction of whole-cell lysate, cells were re-suspended in lysis buffer (150 mM sodium chloride, 1% triton x-100, 50 mM Tris HCl pH8.0 with freshly added proteinase inhibitor (Sigma, P8340)) and incubated on ice for 15 min. The samples were then centrifuged at maximal speed (20,000xg) for 30 min at 4 °C and supernatant was retained. For extraction of histones from nuclei, we followed acid extraction protocol. In brief, cells were resuspended in Triton Extraction Buffer (TEB: PBS containing 0.5% Triton X-100 (v/v), 2 mM phenylmethylsulfonyl fluoride (PMSF), 0.02% (w/v) NaN3), and incubated on ice for 10 min. The samples were then centrifuged at 650xg for 10 min at 4 °C, washed once with TEB and spun down again. Nuclei

were re-suspended in 0.2 N HCl and incubated overnight at 4 °C. The following day, the supernatant was neutralized with 2 M NaOH at 1/10 of the volume of the supernatant. Protein concentration was quantified using the Pierce BCA protein assay kit (Thermo Fisher Scientific, 23227). A total of 25-35 μg proteins from the whole cell extracts and 5-10 μg from nuclei extracts were loaded onto 12.5% or 14.5% Tris-HCl gel for SDS-PAGE analysis. Primary antibodies used: Anti-HA (12CA5) added in 1:1000 dilution, anti-α-GAPDH (Abcam, ab181602) added in 1:5000 dilution, anti-α-Tubulin (Millipore, ABT170) added in 1:1000, anti-H3.3 (Abcam, ab176840) added in 1:1000, anti-H3 (Abcam, ab1791) added in 1:5000, anti-HIRA (Active Motif, clone WC119.2H11) added in 1:1000. Secondary antibodies used: Goat anti-Rabbit IgG (H + L), HRP (Invitrogen, 31460) added in 1:10,000, and Goat anti-Mouse IgG (H + L), HRP (Invitrogen, 31430) added in 1:10000. SuperSignal™ West Pico PLUS Chemiluminescent Substrate (Thermo, 34580) was added for signal detection, and the membrane was imaged by ChemiDocTM MP Imaging System (BioRad). In between antibodies, membrane was stripped using the stripping buffer (2% SDS, 0.0625 M Tris-HCl pH6.8, 0.8% β-mercaptoethanol) for 45 min with agitations and later washed for 3 to 5 times in PBS-T.

## Quantitative real-time PCR

RNA was isolated using the Direct-zol RNA Miniprep Kit (Zymo Research cat.no. R2052) according to the provider's instructions. Between 0.5-2 μg of RNA was further used for reverse transcription with the High-Capacity cDNA Reverse Transcription Kit (Applied Biosystems cat.no. 4368814). Quantitative PCR was performed in triplicates or quadruplicate using Fast SYBR™ Green Master Mix (Applied Biosystems cat.no. 4385610) on a QuantStudio™ 5 Real-Time platform (Applied Biosystems cat.no. A34322). Relative mean expression was normalized to the expression levels of GAPDH or β-Actin. To quantify expression levels of canonical histones, we designed primer sets common to all H3.1 or H2B genes.

## Generation of HIRA KO cell lines

To generate HIRA KO, we utilized CRIPR/Cas9 genome editing system with single guide RNAs (sgRNAs) cloned into SpCas9 px330 plasmid (Addgene, #98750). Three px330-mCherry-sgRNAs vectors were co-transfected into H3.1 and H3.3 sensor cell lines using TransIT-X2 Transfection Reagent (Mirus Bio, MIR6003) as specified by provider's instructions. Two of px330 vectors expressing sgRNAs were designed to remove coding sequence of exon 7 whereby third sgRNA was used for targeted mutagenesis of exon 4. Cells were sorted 48 h post-transfection for mCherry signal and single clones were picked up for further validation. To validate excision of exon7, clones were grown on gelatin-coated plate and genomic DNA was extracted with lysis buffer (100 mM Tris pH8.0, 5 mM EDTA, 0.2% SDS, 200 mM NaCl, 0.2 mg/ml Proteinase-K). Extracted DNA was used as a template for PCR reaction with external primers flanking deletion area, and internal primers.

## ChIP-seq

mESCs were cross-linked with formaldehyde (37% stock, Baker, cat. no. 7040.1000) to final concentration of 1% for 5 min at room temperature, and quenched by adding 0.125 M of freshly prepared glycine (BioLab, cat.no.000713239100) for additional 5 min. Cells were collected by scraping and centrifuged at 500xg for 5 min at 4 °C. Pellet was washed in ice-cold PBS supplemented with cOmplete EDTA-free protease-inhibitor cocktail (04693132001, Roche), spun down and re-suspended for a second time in ice-cold PBS with protease-inhibitor. Volume was then split into LoBind Eppendorf tubes to get 10^7 cells/tube. After centrifugation, cells were snap-frozen in liquid nitrogen and stored at −80 °C.

Pellet was re-suspended in 20 ul of cell lysis buffer (50 mM Tris pH8.0, 150 mM NaCl, 1% TritonX-100, 0.1% sodium deoxycholate, 5 mM CaCl2, EDTA-free protease inhibitor cocktail) and incubated on

ice for 15 min. Cells were centrifuged at 84 5× g for 10 min at 4 °C. The supernatant was discarded, and pellet was re-suspended in 10ul of lysis buffer supplemented with 17.5 units of micrococcal nuclease (Worthington MNase, LS004798) and chilled on ice for 10 min. Samples were incubated at 37 °C for 15 min. To stop reaction, 20 mM EDTA was added and samples were vortexed and placed on ice for 30 min. Cell lysate was then centrifuged at max speed (20,000xg) for 10 min at 4 °C and supernatant moved to a separate tube. Reaction volume was increased to 180ul with RIPA buffer (10 mM Tris pH8.0, 140 mM NaCl, 1 mM EDTA, 0.1%SDS, 0.1% sodium deoxycholate, 1% Triton X-100, EDTA-free protease inhibitor cocktail), and reaction was then split into separate wells of 96-well LoBind Eppendorf plate. ~5 µg of anti-HA (12CA5), ~5 µg of anti-myc (9E10), ~3 µg of H2B (Abcam, ab1790) and ~3 µg of anti-H3K27ac (Abcam, ab4729) antibodies was added to the separate wells and incubated at 4 °C for 2 h with rotations. Anti-HA and anti-myc antibody stocks were the supernatant of the respective hybridoma cell cultures grown in miniPERM bioreactors in-house by the Weizmann Institute Core Facility Antibody Unit. All subsequent steps were carried out as described in ref. 105. In brief, 20 µl of protein G dynabeads were added to each sample and incubated with gentle tumbling for one more hour at 4 °C. Samples were magnetized and washed: 6x RIPA buffer, 3x RIPA 500 containing 500 mM NaCl (supplemented with EDTA-free protease inhibitor cocktail), 3x LiCl buffer (10 mM Tris pH8.0, 0.25 M LiCl, 0.5% NP-40, 0.5% Sodium Deoxycholate, 1 mM EDTA, EDTA-free protease inhibitor cocktail) and 3x 10 mM Tris pH7.5 (supplemented with EDTA-free protease inhibitor cocktail). Samples were resuspended in 10 µl of 10 mM Tris pH7.5, and subjected to End repair reaction containing 13.58 µl of end repair buffer (100 mM Tris pH7.5, 20 mM MgCl2, 20 mM DTT, 2 mM ATP, 0.8 mM each dATP, dCTP, dGTP, dTTP), 1.25 µl of T4 polynucleotide kinase (NEB cat#M0201) and 0.167 µl of T4 polymerase (NEB cat#M0203). The samples were incubated 22 min at 12 °C followed by 22 min at 25 °C. After incubation, the samples were magnetized, washed with 150 µl of 10 mM Tris pH8.0 and resuspended in 20 µl of 10 mM Tris pH 8.0. Resuspended samples were subjected to A-tailing reaction containing 8.5 µl of A-base buffer (10 mM Tris pH 8.0, 10 mM MgCl2, 50 mM NaCl, 1 mM DTT, 0.58 mM dATP) and 1.5 µl of Klenow fragment 3′ → 5′ exo- (NEB cat#M0212). Incubation was performed at 37 °C for 30 min. The samples were then magnetized, washed with 150 µl of 10 mM Tris pH8.0 and resuspended in 9 µl of 10 mM Tris pH 8.0. 2.5 µl of indexed oligo adapters were added to each sample in a ligation reaction using the Quick ligase Kit (NEB cat#M2200) and incubated at 25 °C for 45 min. After incubation, beads were magnetized and washed 3x with 150 µl of RIPA buffer. The beads were then resuspended in 24 µl of chromatin elution buffer (10 mM Tris pH 8.0, 5 mM EDTA, 300 mM NaCl, 0.6% SDS) supplemented with 1 µl of 0.56 µg/µl RNase A and incubated for 30 min at 37 °C. 22.5 µl of chromatin elution buffer supplemented with 2.5 µl of proteinase K (20 units/µl) was later added and the samples were incubated for additional 2 h at 37 °C and for 12–16 h at 65 °C. Tagged DNA products were then cleaned using 2x SPRI beads, and PCR amplified using the KAPA HiFi HotStart Ready Mix kit (Kapa Biosystems cat#KK2601) with 14 PCR cycles. Finally, DNA libraries were cleaned with 0.8× SPRI beads, pooled and sequenced on Illumina NextSeq 500/550 or NovaSeq 6000, with the parameters 51/51 or 61/61 for R1/R2 on NextSeq and NovaSeq, respectively. Demultiplexing and base calling of reads was performed with bcl2fastq. The number of aligned reads for each sample is given in the Supplementary Table 2.

### ES-Blastocyst injection and generation of transgenic mice
To generate chimeric mice, mouse embryonic stem cells bearing cleavable version of H3.3-sensor or H3.1-sensor were injected into (C57BL/6xDBA) B6D2F1 host blastocyst (Envigo), harvested after hormone priming of 3–4 weeks old B6D2F1 females by intraperitoneal injection of pregnant mare serum gonadotropin (PMSG, Vetmarket)

and followed by an injection of human chorionic gonadotropin (hCG, Sigma) 46 hr later.

For germline transmission, male chimera mice were mated with C57BL/6 females and progeny was genotyped for transgenic alleles by PCR with PRIMERS12-14. Expected fragment size of a wt allele is ~350 bp and of mutant is ~200 bp. Male and female offspring (of the C57BL/6 J background) carrying the sensor allele were further bred to obtain homozygous mice. All animals were given free access to food and water and were maintained under controlled conditions with 12 hr light–dark cycle at 22 °C degrees (±2 °C) and 55% humidity (±10%). Breeding experiments were performed on mice that were 8–12 weeks old. Mice were handled according to the Animal Protection Guidelines of Weizmann Institute of Science, Rehovot, Israel and experiments were approved by relevant Weizmann Institute IACUC (#08241020-2 and #02610320-2).

### Isolation of hepatocytes
For extraction of hepatocytes, two adult mice (14 weeks old) carrying either H3.1- or H3.3- sensor were sacrificed. Mice were anaesthetized with 100 mg/kg Ketamine (Zoetis Manufacturing & Research) and 10 mg/kg Xylazine (EuroVet) dissolved in 1×PBS was injected intraperitoneally. Following anesthesia, livers were perfused as previously described[106], with some adjustments. First, a 27 G syringe was connected to perfusion line and pump inserted into the *vena cava*. Next, 7-10 ml of EGTA buffer solution, pre-warmed to 37 °C, was perfused to wash away the blood, continued with perfusion by 12-20 ml of enzyme buffer solution (EBS), supplemented with 2.3U of Liberase (Roche) and pre-warmed to 37 °C. Perfused livers were explanted into a Petri dish, containing 25 ml of pre-warmed EBS and gently minced using forceps. Liver cells were filtered through a 100µm cell strainer, and spun down at 30 g for 3 min at 4 °C to obtain the hepatocyte-enriched pellet. After centrifugation, supernatant was aspirated and the pellet was resuspended in 25 ml of cold EBS. To enrich for live hepatocytes, 22.5 ml Percoll (Sigma-Aldrich) mixed with 2.5 ml of 10×PBS was added to the cells, and cells were centrifuged at 34xg for 10 min at RT. The supernatant containing the dead cells was then aspirated and cells were resuspended in 1xPBS, fixated as described above and stored in −80 °C.

### Isolation of mouse embryonic fibroblasts
To culture MEFs, female mouse carrying H3.1-sensor was mated with a male carrying H3.1-sensor and sacrificed at day 12.5 *post coitum*. Embryos were harvested and dissected in ice-cold 1xPBS, followed by removal of internal organs and the head. Next, embryos were transferred into Eppendorf tubes with trypsin for 5 min at 37 °C. Cells were plated on 10 cm plates in MEF medium, containing DMEM (Gibco, 41965-039), 20% fetal bovine serum (Biological Industries, 04-001-1 A), 0.1 mM beta-mercaptoethanol (Gibco, 31350-010), penicillin/streptomycin (Biological Industries, 03-031-1B), 1 mM L-glutamine (Biological Industries, 03-020-1B), and 1% nonessential amino acids (Biological Industries, 01-340-1B). MEFs were grown in incubators in hypoxia with 5% levels of O2.

### ChIP-seq data processing
Paired-end reads were aligned to the mm10 genome assembly downloaded from the UCSC browser https://hgdownload.soe.ucsc.edu/goldenPath/mm10/bigZips/latest. Bowtie2 was used for alignment with default parameters, and sorted using samtools. Aligned files were then imported into R with readGAlignments function from GenomicAlignments package[107], requiring minimum mapping quality of 10. Mapping quality was specified with ScanBamParam function from Rsamtools R package[108]. Aligned reads are then converted into GRanges object using GenomicRanges package[107]. Fragment sizes >300 bp were filtered out and 5′-ends of reads were then shifted by 80 bp coinciding to the half of average fragment length. Mean read counts over genomic regions were calculated using ScoreMatrixBin function

from R package genomation[109] and normalized so that the average normalized sequencing depth is -0.4

## Used datasets

Coordinates of transcripts for mouse GRCm38.p6 genome assembly were obtained from the TxDb.Mmusculus.UCSC.mm10.knownGene Bioconductor library, whereby selecting only the longest transcript of each gene. To unambiguously distinguish between TSSs and TESs, transcripts <2000 bp were discarded. Finally, only transcripts overlapping RNA data seq in ref. 45 were used for the analysis (total of 19592 transcripts). Promoters were defined as regions 1 kb upstream of TSSs. chromHMM defined genomic coordinates of enhancers, active promoters and transcriptional elongation used in Fig. 2c and Supplementary Figs. 1f, 3c, and 3e were downloaded from https://github.com/guifengwei/ChromHMM_mESC_mm10. Given the low mappability and genomic coverage over heterochromatin, we analyzed it separately: first, H3K9me3 peaks, used as a proxy of constitutive heterochromatin regions, were obtained from https://github.com/elsasserlab/publicchip[31,81] whereby selecting only regions of 150-2000 bp long. Next, top 2000 peaks were filtered based on the highest MNase read density. High-coverage MNase dataset[77] was used to select heterochromatin regions of an appropriate read coverage. Coordinates of compartment A and B were defined previously[67]. Methylation data in mES cells used in Supplementary Fig. 2b was taken from[110]. For Fig. 1j, k and Supplementary Fig. 2d all annotated enhancers (active, primed and poised) were downloaded from ref. 45. In Figs. 4g, 5a, and 5e–h and Supplementary Fig. 7a, b only active and primed enhancers were used[45]. HIRA peaks used in Fig. 5b were downloaded from ref. 42.

For peak calling of H3K27ac in Fig. 1k and Supplementary Fig. 8b, c, MACS[111] was used with default parameters and appropriate input file. In Supplementary Fig. 8b, only high confidence H3K27ac peaks that were common between two datasets were used. In cases when coordinates were provided in mm9 annotation, liftOver function from the rtracklayer package[112] was used to convert them into mm10 assembly. The chain file used for liftOver conversion was downloaded from

http://hgdownload.soe.ucsc.edu/goldenPath/mm9/liftOver/mm9ToMm10.over.chain.gz.

RNA dataset used for analysis of hepatocytes was downloaded from ref. 113 and RNA expression levels of MEF was downloaded from GSE153578[114].

## Genomic analysis of ChIP-seq data

Exchange rates over genomic positions were calculated as $\log2 (myc +0.05) - \log2 (HA + 0.05)$, where 0.05 was added to decrease noise level of regions with low read counts. For comparison of histone exchange and histone modifications, enrichment was calculated as $(\log2_{mark} + 1) - (\log2_{histone\_input} + 1)$. When comparing exchange rates to expression levels, RNA datasets were taken from refs. 45, 113, 114 ($\log2 + 1$ of normalized RNA levels). In Fig. 1g, genes were ranked by RNA expression levels into equally-sized bins whereby bins with low MNase coverage[77] were filtered out. Mean exchange and expression levels were calculated for each bin. To compare exchange rate to external dataset, estimated turnover index was taken from ref. 26. In Fig. 1k, binning of H3K27ac peaks was performed based on H3K27ac peak height. Mean exchange level was compared to the mean H3K27ac and RNA expression of the closest gene groups.

In Fig. 2c, mean exchange within defined genomic segments was normalized by subtracting the mean exchange of a control region, defined by 20 kb shifting of analyzed segments.

In Fig. 2d (upper panel) genes were binned into four groups based on RNA expression levels (from lowest to highest). Mean turnover was calculated for each bin surrounding 2 kb window around TSSs and visualized as meta-profiles.

In Fig. 2d lower panel: each gene is extended 5 kb upstream and downstream from its TSS and TES, respectively. Flanking regions were

then divided into 10 500bp-segments. Similarly, coding sequence of each gene was divided into a 100 segments. For each segment, myc and HA read coverage was calculated separately. Next, genes were grouped into gene collections based on their expression levels. Exchange rate was calculated for each gene group and represented as rows in heatmap.

In Figs. 1g–i, 1k, 6b, c, 6e and Supplementary Fig. 1h a trendline was calculated using Friedman's super smoother function.

## Analysis of Polycomb binding

For comparison of histone exchange and Polycomb binding at gene promoters, reads of Polycomb components were log2 transformed as log2(Polycomb+1), as input was not available for each dataset. To compare exchange levels to the PRC2 binding levels, bivalent promoters are ordered and binned by the PRC2 enrichment score, calculated as the read sum of SUZ12, EZH2 and JARID2.

## Analysis of repetitive sequences

Raw reads were mapped to a dataset of murine repeat sequences downloaded from Repbase (http://www.girinst.org/repbase/), using bowtie2 with standard parameters. Aligned reads were then imported into R without filtering for mapping quality. Read counts per repetitive sequence were normalized by dividing to the total number of aligned reads genome-wide. Sequences with <100 number of aligned reads per sample were discarded as in ref. 31. Additionally, rRNA sequences were filtered out. To identify elements enriched for H3.3, log2-fold enrichment over input (for Elsässer dataset) or H3.1 (for our dataset) was calculated.

To profile read density over repetitive sequences in Fig. 4e, reads mapping to multiple positions were assigned to one of the best matches whereby with enough coverage, read counts were effectively averaged for multiple instances. Aligned reads were imported into R without filtering for mapping quality. Read counts were calculated over 250 windows covering full-length IAP elements whose coordinates were downloaded from https://github.com/elsasserlab/publicchip[81]. For Fig. 4f, g only reads with mapping quality >10 were retained. For ATAC-seq analysis, fragments <120 bp were discarded, and the number of reads was counted over the active and primed enhancer elements from ref. 45 as well as IAP elements sorted and binned by decreasing levels of H3.3. Only 5' ends of reads were used for counting. Mean values were then calculated for each bin and color scale of the heatmap was adjusted to represent minimum and maximum read counts over enhancers and IAPs.

## CTCF sites

CTCF sites containing motif orientation were downloaded from ref. 115. For Fig. 3d–g and Supplementary Fig. 4b motif orientation of CTCF sites were taken into account: when calculating read counts with ScoreMatrixBin function, strand.aware = T argument was used. To select CTCF sites with asymmetric exchange, exchange differences were calculated between the first two nucleosomes downstream and the first two nucleosomes upstream of the CTCF sites. CTCF sites with the highest top 10% exchange differences were selected. Read counts of histone marks were then calculated over these sites. Log2-fold enrichment of histone marks was calculated over appropriate input files as $(\log2_{histone\_mark} + 1) - (\log2_{input} + 1)$ and further normalized by subtracting a log2-fold enrichment within control regions represented by bottom 10% of asymmetrically exchanged CTCF sites. For Fig. 3g mean exchange was calculated for four nucleosomes surrounding CTCF sites overlapping compartment A or B as annotated from ref. 67 and subtracted from mean exchange of control region shifted 20 kb downstream.

For Fig. 3e, CTCF cis-regulatory sites were downloaded from AnnotationHub R package, using the following code:

```
ah <- AnnotationHub()
```

query_data <- subset(ah, preparerclass == "CTCF")

subset(query_data, species == "Mus musculus" & genome == "mm10" & dataprovider == "JASPAR 2022")

CTCF_mm10_all <- query_data[["AH95568"]]

CTCF motifs were overlayed with ChIP-seq data on CTCF in mESC from ref. [115] to determine CTCF-occupied and unoccupied sites. Exchange was next calculated for these two categories but only for four nucleosomes immediately surrounding CTCF motifs (two nucleosomes upstream and two nucleosomes downstream of CTCF motif). These are chosen due to the highest rate of exchange observed in Fig. [3]d. In total 31 bins were covering these nucleosomes, and for each bin, mean myc and HA signals over 25504 (CTCF-occupied), 15830 (CTCF-unoccupied) and 53905 (Shifted) regions were calculated first and used for calculating exchange rates as (log2 $mean_{myc} + 0.05)-(log2 \, mean_{HA} + 0.05)$.

### HIRA analysis

To assess effect of HIRA and thus H3.3 depletion on histone exchange at open chromatin, mean exchange (myc/HA, log2) was calculated within active and primed enhancers as well as promoters overlapping H3.3 peaks from refs. [31], [81]. Mean exchange was then subtracted by mean exchange over active and primed enhancers and promoters that do not overlap with H3.3 peaks. As for heterochromatin, mean exchange was first calculated for H3.3-H3K9me3 overlapping peaks, and subtracted by mean exchange over H3K9me3 only peaks as defined by refs. [31], [81]. Student *t*-test was performed on the biological replicates to determine statistical significance between observed and expected exchange levels of WT and HIRA$^{-/-}$ cell lines. *P*-values < 0.05 were considered significant.

### Methylation analysis

Number of CGs and mean methylation levels were calculated for promoter regions. Occupied promoters (log2(HA + 0.05)) were then stratified based on turnover level into high and low-turning-over based on following criteria:

High turnover = log2(HA + 0.05) > 1 & log2(myc+0.05)-log2(HA + 0.05) > 1.55

Low turnover = log2(HA + 0.05) > 1 & log2(myc+0.05)-log2(HA + 0.05) < −0.45

(see also Supplementary Fig. 2a). Methylation data was downloaded from GSE30206.

### Statistics and reproducibility

Statistical significance was determined by Student's two-tailed *t*-test. Methods for statistical tests, the exact value of n, and definition of error bars are indicated in figure legends. *P*-values < 0.05 were considered significant. ChIP-seq experiments on cell lines expressing cleavable form of exchange sensor have at least two replicates, in line with a common practice in the genomic filed. Sequencing libraries that did not match quality-control measures, showing low internal correlations to respective repeats were not considered for the analysis. ChIP-sequencing for H3.1-cleavable sensor was performed three times, whereby one of the replicate was excluded from the analysis, due to lower correlation of HA signal to HA signal of other two replicates. Genome-studies were performed on individual replicates to verify conclusions for each figure, but for the final versions of figures replicates were pooled together. In case when one of replicates was sequenced to a higher depth, pooling was performed after downsampling of a replicate with higher coverage prior to pooling such that it matches sequencing depth of less sequenced replicate. ChIP-seq experiments involving HIRA knock-out were performed on two independent cell lines (for both, H3.1-tagged and H3.3-tagged samples) with different HIRA genotypes. ChIP-seq of non-cleavable version was representing a control experiment used mostly to validate specificity of anti-myc and anti-HA antibodies, and therefore one sample per group was sufficient for the analysis. Additionally, one sample was used for profiling native H2B signal in cells expressing cleavable sensor, as a control to distribution of tagged H2B variant. Similarly, only one sample derived from in vivo tissues were used for sequencing, as intention was to demonstrate utility of the system for in vivo application, and therefore one sample was sufficient. Exception to this was H3K27ac that was profiled on two hepatocytes samples, isolated from either H3.1- or H3.3-sensor carrying mice. RT-qPCR experiments were performed in three or four technical replicates. The experiments were not randomized and the Investigators were not blinded to allocation during experiments and outcome assessment.

### Reporting summary

Further information on research design is available in the Nature Portfolio Reporting Summary linked to this article.

## Data availability

ChIP-seq data have been deposited in the GEO under accession number GSE213076. The following published ChIP-seq datasets were used in this study: H3K4me3 (ref., GSM723017), H3K4me3 input (GSM723020), H3K27me3 (GSM747539, GSM747540, GSM747541), H3K27ac (GSM1891651, GSM1891652), H3K36me3 (GSM801982, GSM801983), H3K4me1 (GSM747542), Polymerase II (GSM747547, GSM747548), H3K79me2 (GSM1526289), H3K79me2 input (GSM1526285), RING1B (GSM2533855), SUZ12 (GSM1199188), EZH2 (GSM1199182), JARID2 (GSM491760), input for H3K4me1, H3K27ac, H3K27me3, H3K36me3 and Polymerase II (GSM747545, GSM747546), H3K9me3 (GSM1375155), H3K9me3 input (GSM1251941), H3.3 (GSM1555116), H3.3 input (GSM1429923), ATAC-seq (GSM3058311, GSM3058312), MNase (GSM3058339, GSM3058340), H3K4me3 of hepatocytes (GSM6597072), H3K4me3 input of hepatocytes (GSM6597078), H3K4me3 of MEFs (GSM769029), H3K4me3 input of MEFs (GSM769030). Source data are provided with this paper.

## Code availability

Data analysis was performed using custom R scripts and publicly available packages and softwares. All codes generated in this study are available upon request.

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

## Acknowledgements

Y.S. is the Louis and Ida Rich Career Development Chair and a member of the European Molecular Biology Organization (EMBO) Young Investigator Program. Research in the Stelzer lab is supported by European Research Council (ERC_StG 852865), the Israel Cancer Research Fund (ICRF), Helen and Martin Kimmel Stem Cell Institute, Yeda-Sela Center, ISF (1610/18), the Schwartz Reisman Collaborative Science Program, and Abisch Frenkel Foundation. This research was also generously supported by Barry and Janet Lang, Hadar Impact Fund, Lord Sieff of Brimpton Memorial Fund, Janet and Steven Anixter, JoAnne Silva, Maurice and Vivienne Wohl Biology Endowment, and Lester and Edward Anixter Family. We are thankful to Naama Meller and Shiran Bar for critical reading of the manuscript, Prof. Shalev Itzkovitz and Dr. Keren Bahar Halpern for their help in isolating hepatocytes, all members of the Stelzer lab for stimulating discussion, and Hernan Rubinstein for graphic work.

## Author contributions

M.D., F.J., G.Y., N.B. and Y.S. conceived and designed the experiments. M.D. carried out the experiments. M.D. and F.J. analyzed the data. G.Y. contributed to ChIP-seq experiments. Y.M. contributed to experimental design. R.M. and Y.R. assisted with the generation of cell lines. A.-H.O. and Y.M. performed chimera injections. S.C. helped with the experiments. R.M. assisted with isolation and ChIP-seq experiment of hepatocytes. M.D. prepared the figures. Y.S. supervised the study. M.D., F.J., G.Y., N.B. and Y.S. wrote the manuscript with input from all the authors.

## Competing interests

The authors declare no competing interests.
