## [Peer Review File · Nature Communications]

Reviewers' Comments:

Reviewer #1:

Remarks to the Author:

Here Dunjić, Jonas, and Yaakov et al. used recently developed genetically encoded histone exchange sensors to report on histone occupancy, incorporation, and exchange, and thereby generate a genome-wide view of selected histone dynamics. Unsurprisingly, the authors find that histone subunit exchange often scales with transcription, but also find variant specific exchanges in coding regions, as well as interesting associations between histone subunit exchange with polycomb complex proteins, CTCF, and heterochromatic regions. Overall, this is a great use of a newly developed technology to uncover new insights into chromatin biology in mammalian cells. I had only relatively minor concerns, which are enumerated below.

1. It is surprising that the authors can achieve genome-wide incorporation of their histone subunit sensors using knock-in of the sensors at a single safe harbor site. Can the authors quantify the amounts of their sensor-based subunits relative to native histone subunits in mESCs. The Western blots in figures 1c and 2b are not quantitative. Furthermore, these Western blots do not define any sort of "kinetics" as indicated by the authors – this language should be revised.

2. The authors propose that histone exchange in the vicinity of CTCF sites is mediated by additional factors (such as SNF2H) and is required for the binding of CTCF. Can the authors test this hypothesis by knocking out (or knocking down) these factors? Similarly, what happens if key CTCF sites are mutated using CRISPR-based disruption? Finally, the use of inducibly degradable CTCF systems (see PMIDs: 28525758, 34429148, 34453048) could clarify how CTCF levels alter exchange rates independently of other cofactors.

Minor concerns:

3. The use of "in vivo" in the abstract could be misleading. I suggest "in situ" or "in cells".
4. In line 261, the author can simply add the reference after "as previously shown".
5. In line 430 "chaperones" is misspelled.
6. In line 467 "cell" should be plural.
7. Some of the legends use "#XXXX" nomenclature. I suggest using "n = XXXX" instead, to avoid any confusion.

Reviewer #2:

Remarks to the Author:

This manuscript from Dunjic et al expands previous work from the lab to adapt a histone biosensor exchange system to mouse embryonic stem cells. The authors report that histone proteins undergo high exchange at active regulatory elements such as enhancers and promoters. This is in line with many previous studies in the field. They also use their system to assess various other genomic regions, including bivalent promoters, CTCF boundary elements, and endogenous retroelements. Overall, the conceptual insights are limited in scope over what has been shown previously regarding histone dynamics. Beyond that, I have some serious technical concerns about the execution of this system in mammalian cells, and the interpretation of data, outlined below. In short, I think that, at best, this system might measure deposition, but it cannot tell the difference between recycling and nucleosome stability. All things considered, I cannot recommend this work for publication at this time.

1. The manuscript should clarify in the text whether the TEV and sensor tags are N- or C-terminal.
2. I have concerns about the expression of the sensors and TEV-labeled histones with respect to the total pool of histone in the cell. The authors report relative RNA levels for tagged RNA compared to endogenous H3.3 in a supplemental figure, but the salient point is really the contribution to the protein pool. There are antibodies that are specific for H3.3 and H3.1, so this experiment is feasible. The authors do not attempt to address this question for the TEV-labeled H2B, but this is just as important.
3. I do not think the authors have accounted for either the stoichiometry or the kinetics of the system in their modeling. In the example of H2B-TEV and H3.3-HA-myc, H3.3 deposition will be

restricted to genes, regulatory elements, and repeats in ESCs, whereas H2B will be distributed evenly throughout the genome. It seems entirely possible that H3.3-HA-myc cleavage might not occur simply because that molecule of H3.3 is not in proximity to an H2B-TEV. Also, the system depends on both H2B dynamics and H3.3 dynamics, that's to say the dynamics of both the TEV histone and the sensor, and I do not think this issue is addressed.

4. Based on the western blot in Fig. 1c, it seems like the nature of the epitope tag is greatly influencing the production or stability of H3.3, that would then have implications for the turnover measurements that the authors are trying to make. The authors should address why there seems to be much more H3.3 present in cells when the epitope tag cannot be cleaved compared to when it can.

5. Based on the western blots shown in Fig. 1c and 2b, there is no double epitope-tagged histone observed under steady state, and all protein in all cases (i.e., H3.1, H3.3, H2B) carries only an HA tag. I do not understand how myc levels are being determined by ChIP if there is no myc-tagged protein present in these cells under the conditions of the experiment. The one browser track shown in Fig. S2c shows strong myc enrichment at hypomethylated regions, and I am concerned that this myc signal is artifactual due to the open chromatin state. This would be evident as strong promoter bias in the cleavable myc data set.

6. I do not understand how a high myc to HA ratio could be observed, such as that shown in Fig. 1f. Perhaps I do not understand how the data is being processed / represented, but it seems that all detected histone should have an HA tag, while only some histone should have a myc tag. I don't understand how there could be myc/HA > 1.

7. I have concerns about putting the H3.1-HA-myc in the safe harbor locus under control of an EF1a promoter. This will decouple H3.1 expression from its normal restriction to S phase, resulting in expression throughout the cell cycle. Further, H3.1 deposition would become uncoupled from the replication fork, as HIRA and ATRX/DAXX-mediated deposition will be predominant over CAF-1 mediated deposition, and this could influence where their H3.1-HA-myc protein is deposited. Also, this result will be highly dependent upon where the H2B-TEV is deposited, which will likely be at the enhancers and promoters that are experiencing high rates of nucleosome turnover.

8. The authors interpret their results from Fig. 2 as H3.1 being dynamic at repeats while H3.3 is more stable. I don't think that the system can distinguish "stability" from recycling. It's been shown in the literature that H3.3 can be recycled back into active regions, and it is possible that this is also happening at repeat regions.

Minor Points:

1. Colors in pie charts for Fig. S1e and S3c are not well defined.

2. Labels on Fig. 5d and S6c are confusing – looks like the authors made H3.1 and H3.3 KOs, not HIRA KOs.

P2P response to the Reviewers

Reviewer #1 (Remarks to the Author):

Here Dunjić, Jonas, and Yaakov et al. used recently developed genetically encoded histone exchange sensors to report on histone occupancy, incorporation, and exchange, and thereby generate a genome-wide view of selected histone dynamics. Unsurprisingly, the authors find that histone subunit exchange often scales with transcription, but also find variant specific exchanges in coding regions, as well as interesting associations between histone subunit exchange with polycomb complex proteins, CTCF, and heterochromatic regions. Overall, this is a great use of a newly developed technology to uncover new insights into chromatin biology in mammalian cells. I had only relatively minor concerns, which are enumerated below.

We thank the Reviewer for the supportive and constructive comments. Accordingly, we have revised our manuscript and provided details on the remaining concerns below.

Major points:

1. It is surprising that the authors can achieve genome-wide incorporation of their histone subunit sensors using knock-in of the sensors at a single safe harbor site. Can the authors quantify the amounts of their sensor-based subunits relative to native histone subunits in mESCs. The Western blots in figures 1c and 2b are not quantitative.

We thank the Reviewer for raising this point. To try and quantify the relative amounts of sensor subunits from the total histone pool, we used antibodies specific to H3.3, H3.1, and H2B. Notably, we detected the H3.3 sensor at a significantly lower abundance than its endogenous counterpart (see **new Extended Data Fig. 1c** and **p2p_Fig. 1a**). This was inconsistent with quantification at the RNA level that showed (as expected) a modest ~2-6 fold reduced expression of cleavable sensor compared to the native H3.3 histones and ~1.6-4.4 fold lower expression of NC sensor (see **new Extended Data Fig. 1b**).

Unfortunately, we failed to detect modified canonical variants with variant-specific antibodies, even though we readily detected the native subunits (see **p2p_Fig. 1b,c**). We also tried two different pan-H3 polyclonal antibodies, as well as antibodies for histone modifications including H3K27ac, H3K4me3, and H2Bub. All tested antibodies recognized endogenous histones but did not detect tagged variants (see **p2p_Fig. 1d-h**). In contrast, when blotting with the HA or myc antibodies, used in the ChIP experiments, we were able to readily detect these sensors (see **Fig. 2b** in the main text and **p2p_Fig. 2 below**).

Given that the results of qRT-PCR for H3.3 variant were overall consistent with native histone expression, we next set up to quantify relative levels of modified H3.1 and H2B variants at the RNA level. For this, we took advantage of conserved histone sequences and used primer sets common to all H3.1 or H2B genes. The results showed that native histones are expressed at least ~16-fold more than the sensor and 44-fold more than the TEV enzyme (see **Extended Data Fig. 3b**).

Our interpretation from the multiple Western blots and qPCRs is that antibody specificity technically limits the detection of the sensors, which are expressed at much lower levels compared to native variants. Nonetheless, these variants can be readily detected with the ChIP antibodies targeted at high-affinity epitopes. Unfortunately, this precludes direct quantitative comparisons between sensor subunits and native histones in the same protein pool.

We agree with the Reviewer that low abundance might be of concern for genome-wide analysis, especially

of canonical histone variants that are deposited genome-wide. For this reason, ChIP libraries of canonical histone variants were sequenced to higher depth as compared to non-canonical H3.3. Additionally, non-cleavable controls were added to test for genome-wide correspondence of myc and HA tags, reflecting on histone occupancies. Our data show that the sequencing depth was sufficient to cover all genomic regions of interest, as shown in **Extended Data Fig. 1f** and **Extended Data Fig. 3c**. Not less important, genome-wide comparisons between tagged H2B histone and its native counterpart showed high levels of correlation (see **Extended Data Fig. 4a**).

Together, while the relatively lower abundance of sensor-based subunits precludes direct quantification of our system compared to the native histones – we don't think this jeopardizes the utility of our system, which clearly confirms previous variant-specific genomic trends and is compatible with the generation of transgenic reporter mice *in vivo* (see **new Fig. 6**). We note that lower sensor abundance is in fact advantageous as it perturbs the cells less, while still integrating and being cleaved genome-wide.

Furthermore, these Western blots do not define any sort of “kinetics” as indicated by the authors – this language should be revised.

We agree with the Reviewer that this Western blot does not show cleavage ‘kinetics’ of the TEV enzyme. This has now been corrected in the text.

2. The authors propose that histone exchange in the vicinity of CTCF sites is mediated by additional factors (such as SNF2H) and is required for the binding of CTCF. Can the authors test this hypothesis by knocking out (or knocking down) these factors? Similarly, what happens if key CTCF sites are mutated using CRISPR-based disruption? Finally, the the use of inducibly degradable CTCF systems (see PMIDs: 28525758, 34429148, 34453048) could clarify how CTCF levels alter exchange rates independently of other cofactors.

We agree with the Reviewer that it would be fascinating to directly investigate the role of CTCF and associated factors in histone exchange, but the construction of suggested cell lines (e.g. targeting the sensor systems into cells already harboring a degradable CTCF system) is tedious and complex. We therefore believe this is outside the scope of this paper and hope the Reviewer will agree on this point.

Following the Reviewers’ comments, we sought to test the hypothesis that histone dynamics in the vicinity of CTCF may depend on CTCF binding. We therefore divided genome-wide CTCF binding motifs into two categories: CTCF-bound and CTCF-unbound (using ChIP data), and measured mean exchange levels at the four surrounding nucleosomes where we observed the highest exchange (**Fig. 3d**). As shown in new **Fig. 3e** and **p2p_Fig.3 below**, mean exchange levels of unbound CTCF motifs were lower than at shifted regions. Contrary, mean exchange around CTCF bound motifs is significantly higher, implying that the

exchange is associated with CTCF binding. Finally, we tone down the statement proposing regulation of histone exchange in the vicinity of CTCF sites is affected by other *trans*-acting factors¹, as further detailed in the manuscript.

Minor concerns:

3. The use of “*in vivo*” in the abstract could be misleading. I suggest “*in situ*” or “*in cells*”.

We agree with the Reviewer on wrong use of *in vivo* in the previous version of our manuscript. We have now, in fact, generated new *in vivo* data from mice carrying the sensors, highlighting the exciting potential of our system for *in vivo* applications. These new data have been added to the revised version of the manuscript and abstract (see new **Fig. 6** and associated section in text).

4. In line 261, the author can simply add the reference after “as previously shown”.

This has been corrected.

5. In line 430 “chaperones” is misspelled.

This has been corrected.

6. In line 467 “cell” should be plural.

This has been corrected.

7. Some of the legends use “#XXXX” nomenclature. I suggest using “n = XXXX” instead, to avoid any confusion.

We have updated the Figures and Figure legends accordingly.

Reviewer #2 (Remarks to the Author):

This manuscript from Dunjic et al expands previous work from the lab to adapt a histone biosensor exchange system to mouse embryonic stem cells. The authors report that histone proteins undergo high exchange at active regulatory elements such as enhancers and promoters. This is in line with many previous studies in the field. They also use their system to assess various other genomic regions, including bivalent promoters, CTCF boundary elements, and endogenous retroelements. Overall, the conceptual insights are limited in scope over what has been shown previously regarding histone dynamics. Beyond that, I have some serious technical concerns about the execution of this system in mammalian cells, and the interpretation of data, outlined below. In short, I think that, at best, this system might measure deposition, but it cannot tell the difference between recycling and nucleosome stability. All things considered, I cannot recommend this work for publication at this time.

We thank the Reviewer for taking the time to study our data and providing constructive criticism. Following the Reviewers' comments, we added new data and revised the text. Most notably, part of the new data is derived from mice harboring the sensors, exemplifying the system's previously unattainable *in vivo* potential. We would also like to ask the Reviewer to consider the points listed immediately below.

These highlight the case for the novelty and importance of our study, and how the new experimental systems make it possible to go far beyond existing methodologies to study histone exchange *in vivo*.

The key points are:

- We map incorporation and exchange patterns of non-canonical (H3.3) and canonical (H3.1 and H2B) histone variants in mouse embryonic stem cells, representing, to our knowledge, the most comprehensive genome-wide dataset generated to date in mammalian cells. In addition, our data include detailed maps of H3.1 and H2B canonical variants, which were not charted before at such genomic scale.
- These data allow robust comparisons between the histone variants, identifying variant-specific exchange associated with transcription and *cis*-regulatory elements, but also previously unappreciated dynamics in heterochromatin and repeat elements. In particular, we showed that classes of repetitive sequences with high H3.3 occupancy exhibited the highest exchange of H3.1 and H2B histones. This dynamic of canonical variants was unexpected, implying that the exchange is not only restricted to open chromatin and gene regulation.
- Our analysis uncovered a surprising association between the incorporation of H3.3 and exchange dynamics of canonical variants in both enhancer and repeat elements, which we functionally validate by knockout of the histone H3.3-specific chaperone HIRA in cells harboring the sensor system.
- In the revised version of the manuscript, we now demonstrate the vast potential of this system for studying the regulation of histone exchange and its impact on gene expression regulation *in vivo*, which was so far largely inaccessible using existing methodologies (see new **Fig. 6** and **Extended Data Fig. 8**).

We fully agree with the Reviewer that the sensors do not distinguish between nucleosome stability and recycling and have rephrased these statements in the text so as not to mislead.

1. The manuscript should clarify in the text whether the TEV and sensor tags are N- or C-terminal.

We apologize for not providing this information. The sensor is tagged on the C-terminal end. This has now been added to the manuscript.

2. I have concerns about the expression of the sensors and TEV-labeled histones with respect to the total pool of histone in the cell. The authors report relative RNA levels for tagged RNA compared to endogenous H3.3 in a supplemental figure, but the salient point is really the contribution to the protein pool. There are antibodies that are specific for H3.3 and H3.1, so this experiment is feasible. The authors do not attempt to address this question for the TEV-labeled H2B, but this is just as important.

To try and quantify the relative abundance of sensor subunits with respect to the total histone pool, we used antibodies that are specific to each of the variants. As shown in **p2p_Fig. 1** (please see page 2 of this document), while this analysis readily detected the native subunits, we mostly failed to detect both cleavable and non-cleavable sensors within these samples. Notably, we could detect relatively low protein abundance of the H3.3 variant (see **Extended Data Fig. 1c** and **p2p_Fig. 1a**). This was inconsistent with quantification at the RNA level that showed (as expected) ~2-6 fold reduced expression of the cleavable

sensor compared to the native H3.3 histones and ~1.6-4.4 fold lower expression of NC sensor (see **new Extended Data Fig. 1b**). To quantify expression of TEV-tagged H2B variant compared to its native counterpart, we designed H2B primer set that is common to all native H2B transcripts. The analysis showed that the H2B-TEV is expressed at least 21 fold less compared to native H2B (see **new Extended Data Fig. 1b**). As we failed to detect the canonical sensors with variant-specific antibodies, we also tried two different pan-H3 polyclonal antibodies, as well as antibodies for histone modifications including H3K27ac, H3K4me3, and H2Bub. All tested antibodies recognized endogenous histones but did not detect tagged variants (see **p2p_Fig. 1d-h**).

Given that the results of qRT-PCR for H3.3 variant were overall consistent with the expression of native H3.3 histones, we next set up to quantify relative levels of our modified H3.1 and H2B variants at the RNA level. To this end, we used the primer pair targeting all H2B transcripts, as described above, and another primer pair common to all H3.1 sequences. The analysis showed that the sensor and TEV protease are expressed at least 16 and 44-fold less compared to native variants, respectively (see **Extended Data Fig. 3b**).

Our interpretation from these multiple Western blots and qRT-PCR analysis is that antibody specificity technically limits the detection of the modified canonical histone variants, which are expressed at much lower levels compared to native histones. Nonetheless, these variants can be captured with antibodies targeted at high-affinity epitopes, which are those used in the ChIP experiments. Unfortunately, this precludes direct quantitative comparisons between sensor subunits and native histones in the same protein pool. We now discuss this point in the revised version of the manuscript. As detailed below in point #3, we do not think that the differential expression of sensor and TEV protease is affecting the interpretation of the results given that for all profiled variants we observed high cleavage rates of the cleavable sensor, indicating that sensor and TEV were in close proximity to each other (see **new Fig. 1c** and **Fig. 2b**).

3. I do not think the authors have accounted for either the stoichiometry or the kinetics of the system in their modeling. In the example of H2B-TEV and H3.3-HA-myc, H3.3 deposition will be restricted to genes, regulatory elements, and repeats in ESCs, whereas H2B will be distributed evenly throughout the genome. It seems entirely possible that H3.3-HA-myc cleavage might not occur simply because that molecule of H3.3 is not in proximity to an H2B-TEV.

We thank the Reviewer for this comment. First, we point out that no quantitative modeling was performed in this manuscript, such that we did not attempt to make claims about stoichiometry or the kinetics of the system. We appreciate the concern of the Reviewer regarding the inherent dependency on the proximity of TEV protease to tagged histones. Indeed, we were also concerned about this point. However, several sources of information support the validity of this approach:

- Western blot results with HA antibody suggest high cleavage efficiency, implying that the TEV was indeed in the proximity of sensor-tagged variants within the nucleosomes (see **Fig. 1c** and **Fig. 2b**). Notably, Western blot images show that the vast majority of H3.3-tagged as well as H3.1-tagged sensors are cleaved, supporting the notion that H2B –TEV is distributed genome-wide.
- As stated by the Reviewer, H2B-TEV is expected to be distributed evenly throughout the genome. To support this, we analyzed H2B-HA (from the H2B sensor) as a proxy for the incorporation of tagged

histones. This analysis confirms that H2B-HA levels correlate with the genome-wide distribution of native H2B signal (see **new Extended Data Fig. 4a**), suggesting that all H3.3-tagged histones are in proximity to TEV, regardless of genomic location.

- Next, to confirm that the exchange does not stem from non-uniform H2B-HA levels, we focused the analysis around TSS's in which we identified the highest exchange levels. When sorting these regions by H3.3 myc levels, we observed overall uniform distribution of H2B HA and no correlation between changes in myc to a specific depletion of H2B-HA, implying that the differences in myc cannot be explained by different H2B TEV distributions within these regions (see **new Extended Data Fig. 4b,c**). Similarly when sorting these TSS regions by decreasing H3.1 or H2B myc signals, we also observed overall uniform distribution of complementary subunit, suggesting that also here H3.1 and H2B histones are deposited in the same regions, and therefore high myc signal is a consequence of new histones deposition (see **new Extended Data Fig. 4d,e**). These results have now been added to the revised version of the manuscript.
- Finally, we used H2B TEV protease when profiling both H3.1 and H3.3 exchange. If locally different TEV occupancies were a major source for variation in myc levels, we would expect to observe the same pattern of histone exchange between the variants. However, this was clearly not the case, and we observed marked differential histone exchange patterns between H3.1 and H3.3 (e.g. see **Fig. 2c,d**).

Also, the system depends on both H2B dynamics and H3.3 dynamics, that's to say the dynamics of both the TEV histone and the sensor, and I do not think this issue is addressed.

This concern was previously addressed in yeast, where the system was originally generated². Specifically, the TEV protease was fused to one of two yeast H2B alleles, whereas HA-myc was fused to the other H2B allele. As 2 dimers of H2A-H2B interact only in the context of the nucleosome, the sensor cleavage should similarly report on the residence time of H2B on a nucleosome, as in the case of the original system (H2B-HA-myc and H3-TEV). Indeed, the results of this experiment recapitulated the behavior of H2B sensor combined with H3-TEV. This demonstrates that, at least in yeast, the system is largely independent of TEV dynamics. Importantly, this point is also reflected by the different exchange patterns of H3.3 and H3.1 sensors despite both relying on H2B-TEV, indicating that TEV-cleavage is not the major driver of myc signal.

4. Based on the western blot in Fig. 1c, it seems like the nature of the epitope tag is greatly influencing the production or stability of H3.3, that would then have implications for the turnover measurements that the authors are trying to make. The authors should address why there seems to be much more H3.3 present in cells when the epitope tag cannot be cleaved compared to when it can.

We agree with the Reviewer that the non-quantitative western blot in the previous Fig. 1c could suggest lower levels for the cleavable H3.3 variant. We have repeated this western blot using an improved dedicated HCl-based extraction protocol for positively charged histones and obtained a more robust signal for H3.3 cleaved variant (see **p2p_Fig. 4** below). We regret the inaccuracy in the original submission, and replaced the image in the revised version of the manuscript (see **new Fig. 1c**). However, we note that the protein level of cleaved H3.3 variant is still lower than for non-cleavable sample, as also observed when blotting was performed by anti-H3.3 antibody (see **new Extended Data Fig. 1c**). Lower protein levels of cleaved H3.3 are consistent with lower expression levels in this cell lines as compared to NC cell lines, when measured by the RNA (see **Extended Data Fig. 1b**). Taken together, we do not find indications that cleavage influences the stability of tagged histones. This is based on the new western blots as well as the original western blots for canonical H3.1 and H2B tagged variants (see **Fig. 2b**). Moreover, HA tags of cleavable and non-cleavable H3.3 variant are highly correlated, and histone exchange rates of our data are correlated to exchange rates of previously published data (see **Extended Data Fig. 1g,h**).

p2p_Fig. 4. New western blot images with HA antibody.

5. Based on the western blots shown in Fig. 1c and 2b, there is no double epitope-tagged histone observed under steady state, and all protein in all cases (i.e., H3.1, H3.3, H2B) carries only an HA tag. I do not understand how myc levels are being determined by ChIP if there is no myc-tagged protein present in these cells under the conditions of the experiment. The one browser track shown in Fig. S2c shows strong myc enrichment at hypomethylated regions, and I am concerned that this myc signal is artifactual due to the open chromatin state. This would be evident as strong promoter bias in the cleavable myc data set.

We have indeed also considered this point and thank the Reviewer for pointing it out. We are confident we are recovering myc-tagged sensor for ChIP for the following reasons:

1. The histone timers were initially calibrated in yeast. The goal was to maximize cleavage of the myc-tagged sensor (while still retaining enough input for ChIP signal), thus optimizing the temporal resolution of the rapidly exchanging histones. Two TEV cleavage sites were tested in yeast – wt ('fast') and a point mutant ('slow') with a 30-fold decreased Kcat. The fast cleavage variant was superior in the dynamic range of the myc ChIP data, while the slow was much more readily detected by western blot. Key to this point, the levels of uncleaved sensor left in yeast were barely detectable by western blot with anti-myc, and in fact, nondetectable with anti-HA, while clear myc ChIP signals were robustly detected.

As histone protein sequences are highly conserved between yeast and mice, we expect cleavage dynamics to be similar. To test this, we now generated cell lines carrying a modified 'slow' variant of TEV cleavage site. Cleavage sequences were identical between mice and yeasts, and indeed, we observed that cleavage patterns resembled the one observed in yeasts (see **p2p_Fig. 5a-b** below). Furthermore, after optimizing

the western blot conditions for mice proteins and exposing for longer time period, we consistently see a faint band of uncleaved sensor using anti-myc (see **p2p_Fig. 5c** below). Notably, ‘fast’ variants (referred as ‘cleavable’ in the text) were used for ChIP experiments included in the manuscript.

2. Our data is consistent with myc signal not being an artifact of open chromatin. This is evident from the different patterns obtained by the various histone variants in the same open chromatin regions (e.g see **new Extended Data Fig. 4**), as well as the significant myc signal detected in closed heterochromatin regions.

6. I do not understand how a high myc to HA ratio could be observed, such as that shown in Fig. 1f. Perhaps I do not understand how the data is being processed / represented, but it seems that all detected histone should have an HA tag, while only some histone should have a myc tag. I don't understand how there could be myc/HA >1.

We fully agree with the Reviewer that in **absolute** terms we maximally expect the same levels of myc and HA at sites of replacement. The ChIP signals however cannot be directly compared in absolute terms. Rather, it is necessary to internally normalize each sample, resulting in relative measures. Consequently, relative enrichment of myc at certain genomic regions versus the overall genome can be higher than HA, and vice versa. We now clarify this better in the text.

7. I have concerns about putting the H3.1-HA-myc in the safe harbor locus under control of an EF1a promoter. This will decouple H3.1 expression from its normal restriction to S phase, resulting in expression throughout the cell cycle. Further, H3.1 deposition would become uncoupled from the replication fork, as HIRA and ATRX/DAXX-mediated deposition will be predominant over CAF-1 mediated deposition, and this could influence where their H3.1-HA-myc protein is deposited. Also, this result will be highly dependent upon where the H2B-TEV is deposited, which will likely be at the enhancers and promoters that are experiencing high rates of nucleosome turnover.

This is an excellent point. To address this concern, we sought to measure exchange of H3.3 and H3.1 sensors in non-dividing post-mitotic cells. To this end, we utilized novel transgenic mice models we have

generated, harboring H3.3 or H3.1 sensor. First, we isolated non-dividing hepatocytes from livers of 14-weeks-old reporter mice, followed by sequencing of both HA and myc. As expected, we observed a correlation between H3.3 exchange and gene transcription, but no correlation was observed for H3.1 exchange, which appeared largely not exchanging. We did however detect a robust HA signal for H3.1, implying that labeled histones are deposited in the genome (see **new Fig. 6a,b** and **Extended Data Fig. 8a,b**). To validate that H3.1 sensor indeed exchanges in rapidly dividing cells *in vivo*, we isolated embryonic fibroblasts from embryos at day 12.5 post coitum. In contrast to post-mitotic cells, this analysis identified a strong positive correlation between high gene expression and rapid H3.1 exchange (see **new Fig. 6c,d**).

Together, these results support the notion that ectopic expression of H3.1 in post-mitotic cells does not lead to misincorporation by the H3.3-specific chaperones. In addition, the successful generation of viable reporter mice strongly argues against potential adverse effects caused by the transgene expression, overall supporting the faithfulness of the sensor system. More broadly, we believe these results are of high interest to the community, holding great potential for enhancing our understanding of histone exchange biology *in vivo*. We thank the Reviewer for prompting us toward more ambitious experiments using this system.

8. The authors interpret their results from Fig. 2 as H3.1 being dynamic at repeats while H3.3 is more stable. I don't think that the system can distinguish "stability" from recycling. It's been shown in the literature that H3.3 can be recycled back into active regions, and it is possible that this is also happening at repeat regions.

We thank the Reviewer for pointing out our confusing and inaccurate use of the term "stability". We have now changed it to "occupancy" throughout the text. We agree with the Reviewer that our system cannot distinguish between stability and recycling. This is common to most commonly used methods for studying histone exchange, including pulse-chase methods. A new method could be envisioned to detect such recycling dynamics, but this is not the intention of the currently described exchange sensors.

Minor Points:

1. Colors in pie charts for Fig. S1e and S3c are not well defined.

We thank the Reviewer for pointing this out. Figure legends and figures have now been updated, with color definitions better explained.

2. Labels on Fig. 5d and S6c are confusing – looks like the authors made H3.1 and H3.3 KOs, not HIRA KOs.

We apologize for this. Figures have now been corrected.

References:

1. Barisic, D. Mammalian ISWI and SWI/SNF selectively mediate binding of distinct transcription factors.
2. Yaakov, G., Jonas, F. & Barkai, N. Measurement of histone replacement dynamics with genetically encoded exchange timers in yeast. *Nat Biotechnol* **39**, 1434–1443 (2021).

Reviewers' Comments:

Reviewer #1:

Remarks to the Author:

The authors have addressed my concerns. However, it is worth mentioning that it is unconventional to omit any sort of highlighting or simple tracking in the revised version(s) of a manuscript. The authors failed to do so here, which makes it unnecessarily challenging for reviewers to easily see what has been changed with respect to prior version(s).

Reviewer #2:

Remarks to the Author:

The reviewers have addressed many of my concerns and now add data from a mouse model to begin to dissect histone deposition in dividing and post-mitotic cells. I do still have technical concerns about what I imagine to be very low levels of myc enrichment and then normalizing data sets to this very low level of signal. On a higher level of data interpretation, while the authors acknowledge in the rebuttal that the system cannot distinguish stability and recycling, I don't think they've really clarified this point in the text, which due to their use of the term "exchange" still reads as though nucleosomes are not being turned over at certain regions. Specifically, in their discussion of what is happening to H3 proteins at transcriptionally elongating regions, they lack key reference to published work when they discuss the idea that there are mechanisms in place to support PTM landscapes, PMID 32895554. I think lack of acknowledgment of this work and the idea of recycling, along with the lack of explicit description of the limitations of this system in the text of the manuscript, muddies the waters on what this system can actually reveal about nucleosome dynamics.

P2P response to the Reviewers

Reviewer #1 (Remarks to the Author):

The authors have addressed my concerns. However, it is worth mentioning that it is unconventional to omit any sort of highlighting or simple tracking in the revised version(s) of a manuscript. The authors failed to do so here, which makes it unnecessarily challenging for reviewers to easily see what has been changed with respect to prior version(s).

We thank the Reviewer for the support on this paper and apologize for not providing track changes.

Reviewer #2 (Remarks to the Author):

The reviewers have addressed many of my concerns and now add data from a mouse model to begin to dissect histone deposition in dividing and post-mitotic cells. I do still have technical concerns about what I imagine to be very low levels of myc enrichment and then normalizing data sets to this very low level of signal.

In the previous response, we stated that we “internally normalize each sample”, which we now realize is an ambiguous definition. By internal normalization, we meant that the data is normalized to the library size (sequencing reads), and not to different genomic regions. Thus, we do not normalize data to regions of e.g. low myc signal. This point is now better clarified in the revised version of the manuscript.

On a higher level of data interpretation, while the authors acknowledge in the rebuttal that the system cannot distinguish stability and recycling, I don't think they've really clarified this point in the text, which due to their use of the term “exchange” still reads as though nucleosomes are not being turned over at certain regions.

We have now added clarifications to the main text. Specifically, we use the terms exchange and turnover synonymously. We agree with the Reviewer about the limitations of the system and the inability to distinguish between histone stability and recycling, which we also more clearly now explain in the text.

Specifically, in their discussion of what is happening to H3 proteins at transcriptionally elongating regions, they lack key reference to published work when they discuss the idea that there are mechanisms in place to support PTM landscapes, PMID 32895554. I think lack of acknowledgment of this work and the idea of recycling, along with the lack of explicit description of the limitations of this system in the text of the manuscript, muddies the waters on what this system can actually reveal about nucleosome dynamics.

We thank the Reviewer for pointing this out and have now added this highly relevant reference to the main text also in the context of histone stability: “While the functional significance of this observation remains to be elucidated, it is possible that retention of histone H3 within gene bodies might serve to stabilize an expression-promoting epigenetic landscape such as H3K36me3.” We note that we had initially acknowledged this key reference in our paper, just within a different context.